# Spatiotemporal dynamics of self-organized branching in pancreas-derived organoids

S. Randriamanantsoa [1,2,3,11], A. Papargyriou [3,4,5,6,11], H. C. Maurer [4,6], K. Peschke [4,6], M. Schuster [4], G. Zecchin [1,3], K. Steiger [7], R. Öllinger [4,8], D. Saur [4,8], C. Scheel [5,9], R. Rad [4,8], E. Hannezo [10], M. Reichert [2,3,4,6,8,12] ✉ & A. R. Bausch [1,2,3,12] ✉

The development dynamics and self-organization of glandular branched epithelia is of utmost importance for our understanding of diverse processes ranging from normal tissue growth to the growth of cancerous tissues. Using single primary murine pancreatic ductal adenocarcinoma (PDAC) cells embedded in a collagen matrix and adapted media supplementation, we generate organoids that self-organize into highly branched structures displaying a seamless lumen connecting terminal end buds, replicating in vivo PDAC architecture. We identify distinct morphogenesis phases, each characterized by a unique pattern of cell invasion, matrix deformation, protein expression, and respective molecular dependencies. We propose a minimal theoretical model of a branching and proliferating tissue, capturing the dynamics of the first phases. Observing the interaction of morphogenesis, mechanical environment and gene expression in vitro sets a benchmark for the understanding of self-organization processes governing complex organoid structure formation processes and branching morphogenesis.

The morphogenesis of glandular branched tissues and cancer types derived thereof results in highly branched tubular structures. Despite their prominent complex architecture, little is known of the underlying self-organization processes. A prominent example of such complex tubular structure formation is the pancreas and pancreatic ductal adenocarcinoma (PDAC).

While classical PDAC organoid cultures display varying degrees of differentiation, recapitulate parental tumor subtypes, or display tumor cell plasticity[1] and heterogeneity[2] to a certain degree, they produce mostly spheroids[3], lacking, to date, any degree of organotypic architecture. However, the tubular structure is one of the key features of PDAC[4] and the morphogenetic program that characterizes tumor growth and formation of a hierarchy of branches lined by a continuous lumen, as well as the heterogeneous differentiation related to it, remains elusive. Importantly, it is the structure of the PDAC defining tumor differentiation which characterizes its aggressiveness and, in turn, patient survival[5], that has yet to be recaptured in a model system accessible for ex vivo experiments.

Here, we embed single murine PDAC cells into a collagen matrix, leading to a morphogenic growth process ultimately establishing

[1]Lehrstuhl für Zell Biophysik E27, Physik Department, Technische Universität München, 85748 Garching, Germany. [2]Center for Functional Protein Assemblies, Technische Universität München, 85748 Garching, Germany. [3]Center for Organoid Systems and Tissue Engineering (COS), Technische Universität München, 85748 Garching, Germany. [4]Klinik und Poliklinik für Innere Medizin II, Klinikum rechts der Isar der TUM, 81675 München, Germany. [5]Institute of Stem Cell Research, Helmholtz Zentrum Muenchen, D-85764 Neuherberg, Germany. [6]Translational Pancreatic Cancer Research Center, Medical Clinic and Polyclinic II, Klinikum rechts der Isar, Technische Universität München, 81675 Munich, Germany. [7]Comparative Experimental Pathology, Institute of Pathology, School of Medicine, Technische Universität München, 81675 München, Germany. [8]German Cancer Consortium (DKTK), partner site Munich, 81675 München, Germany. [9]Department of Dermatology, Ruhr-University Bochum, 44791 Bochum, Germany. [10]Institute of Science and Technology Austria, A – 3400 Klosterneuburg, Austria. [11]These authors contributed equally: S. Randriamanantsoa, A. Papargyriou. [12]These authors jointly supervised this work: M. Reichert, A. R. Bausch. ✉e-mail: maximilian.reichert@tum.de; abausch@mytum.de

complex 3D branched structures resembling organotypic architectures. Branched organoid morphogenesis starts with the formation of elongated structures by proliferating cells, followed by a second phase where cells migrate, continuously invade the matrix, and build a tree-like architecture through branching events. In a third phase, branches thicken before entering a fourth phase where micro-lumens nucleate along the structure, eventually coalescing into one single continuous duct. During morphogenesis, organoids undergo epithelial-to-mesenchymal transition (EMT) during branch elongation, and activate an epithelial gene expression program upon maturation. We evidence the critical role of cell proliferation, matrix remodeling, contraction and ion channel flux in the different morphogenic phases. A minimal theoretical model based on the balance between cell invasion, proliferation and branching events, captures the major hallmarks of the developmental branching dynamics. Together, these results highlight the capacity of tumor cells to self-organize in complex structures, and provide an experimental system to study the dynamics of branching morphogenesis and lumen formation in vitro. Our model system paves the way for the investigation of fundamental mechanisms of tumor initiation and invasion, as well as of inherent programs responsible for branching and intra-tumor heterogeneity in pancreatic carcinogenesis at the single-cell level.

## Results

### Extracellular matrix properties drive complex branched phenotypes

Most frequently organoids are cultured inside a protein mixture such as Matrigel[6]. In Matrigel, normal or transformed epithelial cell types, including pancreatic cancer epithelial cells, self-organize into spherical structures with hollow lumen and polarized epithelial lining (Fig. 1a). While distinct epithelial cell types from different organs display minute morphologic differences, organoids derived from the same class of epithelial cell are phenotypically very similar under established culture conditions. Yet, as the tumor microenvironment and physical properties of the ECM are known to be pivotal mediators of tumor progression[7], we investigated here the difference between Matrigel and floating collagen gels (rat tail collagen type I). We used primary tumor cells collected from a genetically engineered mouse model of pancreatic cancer Ptf1a$^{Cre/+}$; Kras$^{G12D/+}$ (KC mouse)[8]. Polymerized collagen gels were detached from the bottom of the culture dish to generate floating gels, adapting a previously described organoid regeneration assay protocol[9]. Indeed, cells cultured using identical media conditions gave rise to noticeably different structures depending on whether they were cultured in a floating collagen matrix or in Matrigel (Fig. 1a–c). Matrigel-grown cells only produced spheroids or cysts with diameters of about 80 μm, as observed with bright field microscopy after 13 days of growth. In stark contrast, we were able to observe branched tubulogenesis, and the formation of both lumen and terminal buds for organoids growing in a collagen I matrix. The extent in length of these complex structures reached up to around 2000 μm within 13 days of growth. Floating collagen organoids also displayed ductal differentiation with cytokeratin-7 (K7) and E-cadherin expression (Fig. 1a). When cells were cultured in a mixture of Matrigel and collagen matrix, organoids were unable to branch even in an equal volume ratio of these two matrix components. Only when providing cells with a matrix containing >70% of collagen, were we able to observe any spheres breaking their symmetry and producing small branching events (Fig. S1a).

Considering the dramatic phenotypic differences of tumor organoids embedded into either Matrigel or floating collagen, we next performed transcriptional profiling of both conditions. Principal component analyses revealed a significant difference in global gene expression (Fig. 1d, e). Surprisingly, we found that tumor organoids cultured within Matrigel are enriched for gene signatures associated with EMT as well as basal-like pancreatic cancer (NES of 1.83, Fig. 1e,

Fig. S2a, b). In contrast, the genetic profiles of the branched organoids in collagen are associated with the classical subtype of PDAC (NES of −1.78) suggesting epithelial differentiation (Fig. 1e, Fig. S2a, b).

To further probe whether the development of branched organoids was a feature of PDAC, we investigated the behavior of tumor cells from a Pdx1$^{Cre/+}$; Kras$^{G12D/+}$; TP53$^{fl/fl}$ mouse (KPC mouse). We report that KPC organoids develop into structures highly similar to organoids of a KC mouse origin, with a three-dimensional thick branched architecture bearing a lumen (Fig. 1f, Fig. S2d, e). On a transcriptional level, compared to KC organoids, KPC organoids displayed enriched signatures for proliferation pathways (Myc targets, E2F targets, etc.) and EMT facilitated by TP53 loss. In stark contrast, we cultured healthy adult pancreatic ductal cells (PDC) in our floating collagen assay, and observed that those cells only gave rise to cyst-like structures (Fig. 1f, Fig. S2c-c').

To investigate whether our system recapitulates tumor morphogenesis in vivo, we next implanted PDAC tumor cells, harvested from a two-dimensional culture, orthotopically into mouse pancreata and allowed cells to engraft and form tumors. Pancreatic tumors were collected after 14 days and processed for histology. As indicated by cytokeratin-19 (K19) immunofluorescence and hematoxylin & eosin staining of single cell-derived branching organoids and corresponding orthotopic tumors, the system indeed displays prominent features of tumor morphogenesis in vivo, including the formation of tubular structures (Fig. 1f, Fig. S1b, c). In summary, we generated an in vitro model system using pancreatic ductal adenocarcinoma cells which recapitulates branching morphogenesis and tubulogenesis, both key features of PDAC. Gene expression profiling of organoids shows a higher degree of epithelial differentiation in branched organoids cultured in collagen, compared to spherical organoids cultured in Matrigel. Furthermore, our results indicate that tumor cells are able to execute inherent self-organizational programs induced by specific biophysical contexts, mirroring tumor morphogenesis to a remarkable degree, both at the transcriptional (Fig. 1d, e) and at the architectural levels (Fig. 1f, Fig. S1b, c).

### Branched organoid morphogenesis displays distinct developmental phases

To shed light on the growth process of these complex architectures, we used live confocal imaging at different time points of growth, over extended time periods of up to 3 days. We observed marked differences in spatial and temporal dynamics at different time points of development, which defined four phases—the onset phase, the extension phase, the thickening phase, and the micro-lumen nucleation and coalescence phase (lumen formation phase)—each characterized by particular patterns of proliferation rate, cell migration and matrix deformation.

**Onset phase.** As the initial single cell proliferates during the establishment phase, newly formed cells arrange into an elongated structure with a main axis of elongation. During this phase, we observed a characteristic back and forth cellular motion in the forming organoid, with intermittent extension and retraction of one- to two- cell-wide branches (Fig. 2a, b and Movie S2, 8). At this stage, proliferation is exponential and homogeneously distributed spatially, as seen with a Ki67 stain, while the organoid major axis length reaches around 500 μm in 5 days (Figs. 1c, 2c–e).

**Extension phase.** A distinct switch in the matrix invasion pattern of the organoid is observable around day 7, in which organoids grow at a rate of around 195 μm/day (Fig. 2a, b, Movie S3, Fig. S5a). At this extension stage, the cellular motion is mostly directed from the core toward the tips of the branches. The cells at the tip of the branches display spiky protrusion as they lead the invasion of the matrix (Fig. 2a, Movie S3, Fig. S1g). The branches´ tips invade the surrounding matrix

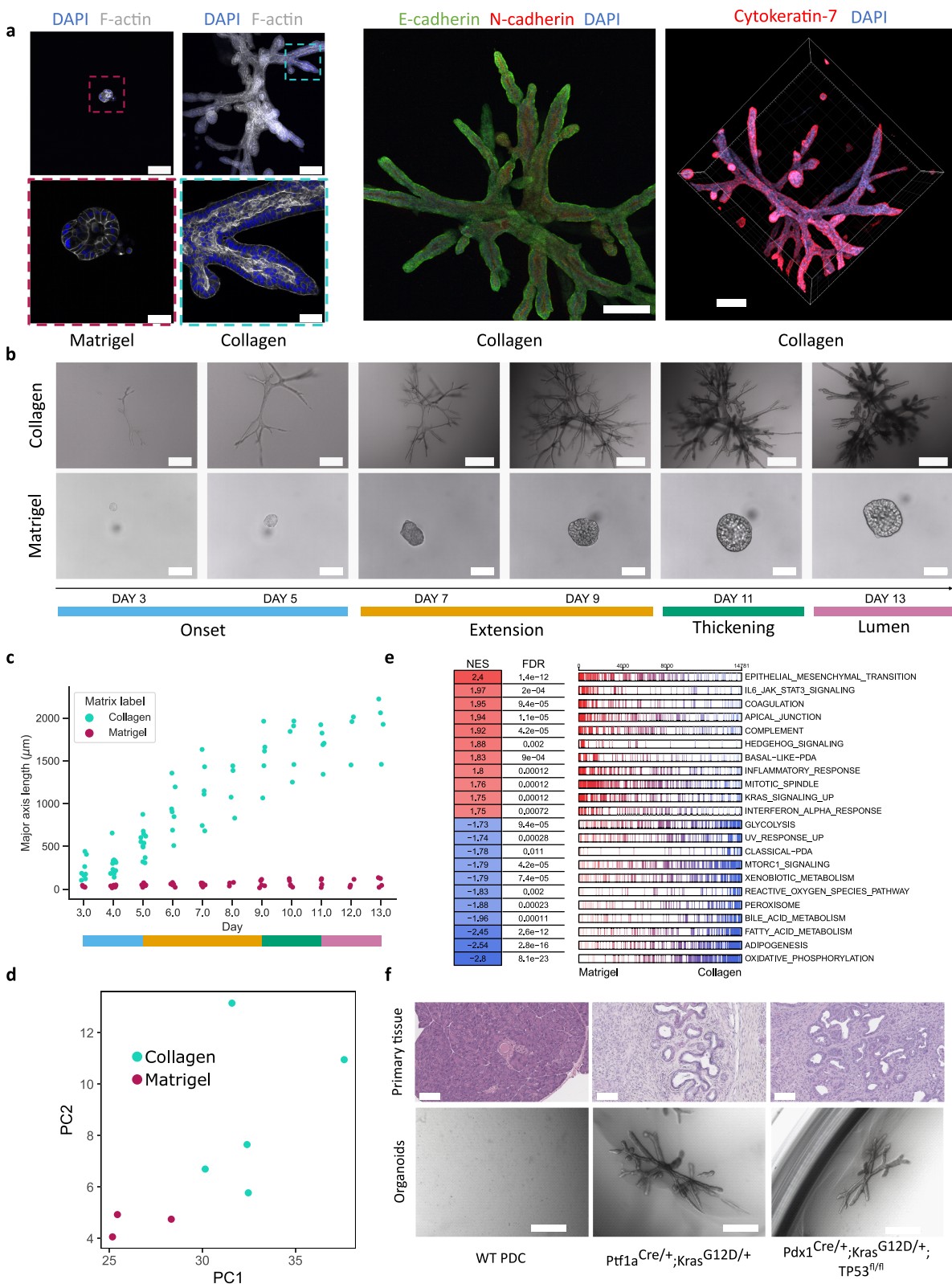

at similar speed across the organoid (Fig. S5b) and cells within the branches' core do not overtake cells at the tip (Fig. S5c, where tracks do not cross over each other). Branching events occur at a rate of around 0.55 times per day at this stage and follow tip cell proliferation events in over 90% of the cases (Fig. 4f, Fig. S5d). In this phase, only a negligible deformation field within the ECM was detectable (Fig. 2b, Movie S9). By reflection microscopy we confirmed a limited formation

of fiber alignment in front of the branches and an accumulation of collagen on the branches' sides (Fig. S1f, Movie S14).

**Thickening phase**. Branching organoids reach a thickening phase after day 9, where branches stop elongating and start widening (Fig. 2a, Movie S4). Contrary to the extension phase, we observed a notable contractile motion across the entire organoid. This is evidenced by a

**Fig. 1 | PDAC cells cultured in floating collagen gels give rise to branched organoids. a** From left to right, dashed boxes outline inset region: Day 13 PDAC organoid in Matrigel staining (blue: DAPI, white: F-actin, $n = 3$ individual experiments), Day 13 PDAC organoid in collagen staining (blue: DAPI, white: F-actin, $n = 3$ individual experiments), E- and N- cadherin staining of a Day 13 PDAC organoid in collagen (green: E-cadherin, red: N-cadherin, blue: DAPI, $n = 3$ individual experiments), 3D reconstruction of a Day 13 PDAC organoid staining in collagen (blue: DAPI, red: Cytokeratin-7, $n = 3$ individual experiments). Scale bars from left to right: 200 μm top and 50 μm bottom, 200 μm top and 50 μm bottom, 100 μm, 200 μm. **b** Bright field snapshots of collagen- (top, $n = 71$ organoids) and Matrigel-grown organoids (bottom, $n = 54$ organoids) at various time points. Colored bars in **b** and **c** indicate the current development phase of collagen-grown organoids: Onset (blue), Extension (orange), Thickening (green), and Lumen Formation phase (pink). Scale bars, top from left to right: 200 μm for the first two pictures, 500 μm for the rest; bottom: 100 μm. **c** Major axis length evolution over time of individual collagen-

(cyan, $n = 71$ organoids) and Matrigel-grown (magenta, $n = 54$) organoids. **d** Principal component analysis of bulk RNA sequencing of Day 13 Matrigel- (magenta, $n = 3$ independent experiments) and collagen-grown (cyan, $n = 5$ independent experiments) organoids. **e** Summary of gene set enrichment analysis between Matrigel- and collagen-grown organoids, showing normalized enrichment scores (NES) and false discovery rates (FDR). Bars represent individual genes for a given gene set. **f** Top, from left to right: haematoxylin and eosin staining of primary tissue sections from a healthy pancreas, a Ptf1a$^{Cre/+}$;Kras$^{G12D/+}$ tumor, and a Pdx1$^{Cre/+}$;Kras$^{G12D/+}$; TP53$^{fl/fl}$ tumor ($n = 3$ technical replicates). Bottom from left to right: bright field images of representative organoids at day 13, grown in collagen from wild type pancreatic ductal cells (WT PDC), Ptf1a$^{Cre/+}$;Kras$^{G12D/+}$ cells, and Pdx1$^{Cre/+}$;Kras$^{G12D/+}$; TP53$^{fl/fl}$ cells, all cultured in PDAC medium (see Methods) ($n = 3$ individual experiments). Scale bars: top, 100 μm; bottom, 500 μm. Fluorescence images shown are confocal slices, except for the 3D reconstruction of the Cytokeratin-7/DAPI staining in **a**.

large contractile deformation field within the ECM, which is strong enough to induce fiber alignment and bundling in the collagen in front of the branches (Figs. 2b, 3a, Fig. S1d), as well as collagen accumulation around the organoid (Fig. 3b, Fig. S1d).

To probe whether the observed deformations and remodeling are elastic—in which case fibers would relax as soon as forces stop being exerted –, or plastic[10]– indicating a permanent deformation of the matrix –, we used Triton-X to degrade the cell membranes, effectively killing the organoids, dissociating them and abolishing the forces exerted on the matrix, leaving the collagen untouched. These cage-like structures remained stable even once organoids were killed by Triton-X treatment, indicating that plastic remodeling of the extracellular matrix occurred during the growth process (Fig. S1d, Movie S14, S15).

As the leading cells in branches retract their invasive protrusions, we observed that branch tips round up and thicken into buds that will later give rise to end-bud structures.

**Lumen formation phase.** Towards the end of the thickening phase and at the beginning of the lumen formation phase, we observed the formation of "fault lines" serving as precursors, where microlumens nucleate, swell and fuse to give rise to a single seamless lumen (Movie S5). We observed extrusion events where cells are ejected from the epithelial layer into the lumen, undergoing apoptosis (Fig. 3c, d, Movie S6). At this point, cells are aligned into a mature and differentiated state (Fig. 5a). Structures strongly express ZO-1 restricted to the apical side facing lumens (Fig. 3c). As a collective effect of the distinct morphogenetic phases, branched organoids form rounded alveoli-like structures at the end of the duct (TEBBOs, Terminal End Bud Branched Organoids). Indicative of the contractile capabilities of the organoids in this phase, we found a strain field within the matrix (Figs. 2b, 3a), and terminal tubular structures expressing alpha-SMA on their basal surface (Fig. 3c). Microlumen formation displays a gradual size increase accompanied by a coalescing process (Fig. 3e, Movie S7). Upon overnight addition of fluorescently labeled Dextran during lumen formation, we observed fluorescent fluid inside the cavity, indicating a degree of permeability of the organoids to their outside environment (Fig. S3c, d).

**A minimal theoretical model captures branching organoid morphogenesis**

We next aimed to probe quantitatively the basic principles governing the extension and the early thickening phase, where the organoids' main tree-like architecture is established, by developing and testing a minimal analytical model. We considered three general processes crucial for shaping a branched organoid: proliferation, branching, and invasion. We consider those parameters in our model as: cellular division at a rate $k_d$, branching processes initiated by tip cells at rate $k_b$ and active migration speed $v_0$ exerted by tip cells during the invasion process. To characterize the morphometrics of a branched

organoid, we denote $w(t)$ the average width of a branch at time $t$, $l(t)$ the average branch length, as well as $N_c(t)$ and $N_b(t)$ respectively the total number of cells and branches in an organoid. For this model, we must write two equations (see SI Note): a simple mass conservation equation and a force balance equation considering potential feedbacks between different parameters. The conservation equation reflects the intuitive idea that different ratios of the growth parameters will give rise to organoids with different morphometrics: smaller active migration speed $v_0$ at constant division rate $k_d$ will tend to give rise to thicker and shorter branches, while increasing the branching rate $k_b$ will increase the number of migrating tip cells, which for a given cell number $N_c(t)$ or division rate $k_d$ will tend to favor thin branches.

For the feedback equation, while during morphogenesis the tip invasion speed $v_0 \approx 80$ μm/day remained roughly constant and independent of other morphogenetic parameters (Fig. S5b), we found that the cellular proliferation rate continuously decreases as a function of time (Fig. 2c, e), reminiscent of findings in Madin-Darby Canine Kidney (MDCK) cells in monolayers arising from a negative mechanical feedback on growth[11,12] (see SI Note for details).

Indeed, we found a strong and consistent negative relationship between branch volume growth, used as a proxy for local proliferation, and branch width (Fig. 4b). This observed decay of the proliferation could be attributed to a contact-inhibition of proliferation processes[13–16], as cells in thicker branches had a larger number of neighbors, as well as the fact that thicker branches need to deform the ECM to a larger degree, thus being under larger stresses than thinner branches. Additionally, other mechanisms have been proposed and could play a role in such negative regulation[17–19]. Remaining agnostic as to its underlying molecular mechanism, this negative regulation of growth was well-fitted by a linear relationship $k_d = k_d^0 \left(1 - \frac{w}{w_0}\right)$, so that we were able to extract the parameters of this feedback mechanism. We then asked whether our model could recapitulate the full time course of organoid branching growth:

$$\begin{cases} \frac{d(N_b l)}{dt} = v_0 e^{k_b t} & (1) \\ \frac{dN_c}{dt} = k_d^0 \left(1 - \frac{w}{w_0}\right) N_c & (2) \end{cases}$$

where the first equation describes the increase in total length of the organoid, which is proportional both to the velocity of individual tips $v_0$ and the total number of tips $e^{k_b t}$ (starting with a single tip at $t = 0$, and given that branching/elongation speed were found to be near-constant), while the second equation describes the increase in cell numbers with the proliferation feedback (Fig. 4b). Simulating the model predicted two phases: a first phase of fast exponential growth at rate $k_d \approx k_d^0$, where ductal thickness remains small so that division is near-maximal, followed by a second phase where proliferation catches up to migration, so that thickness reaches a plateau value

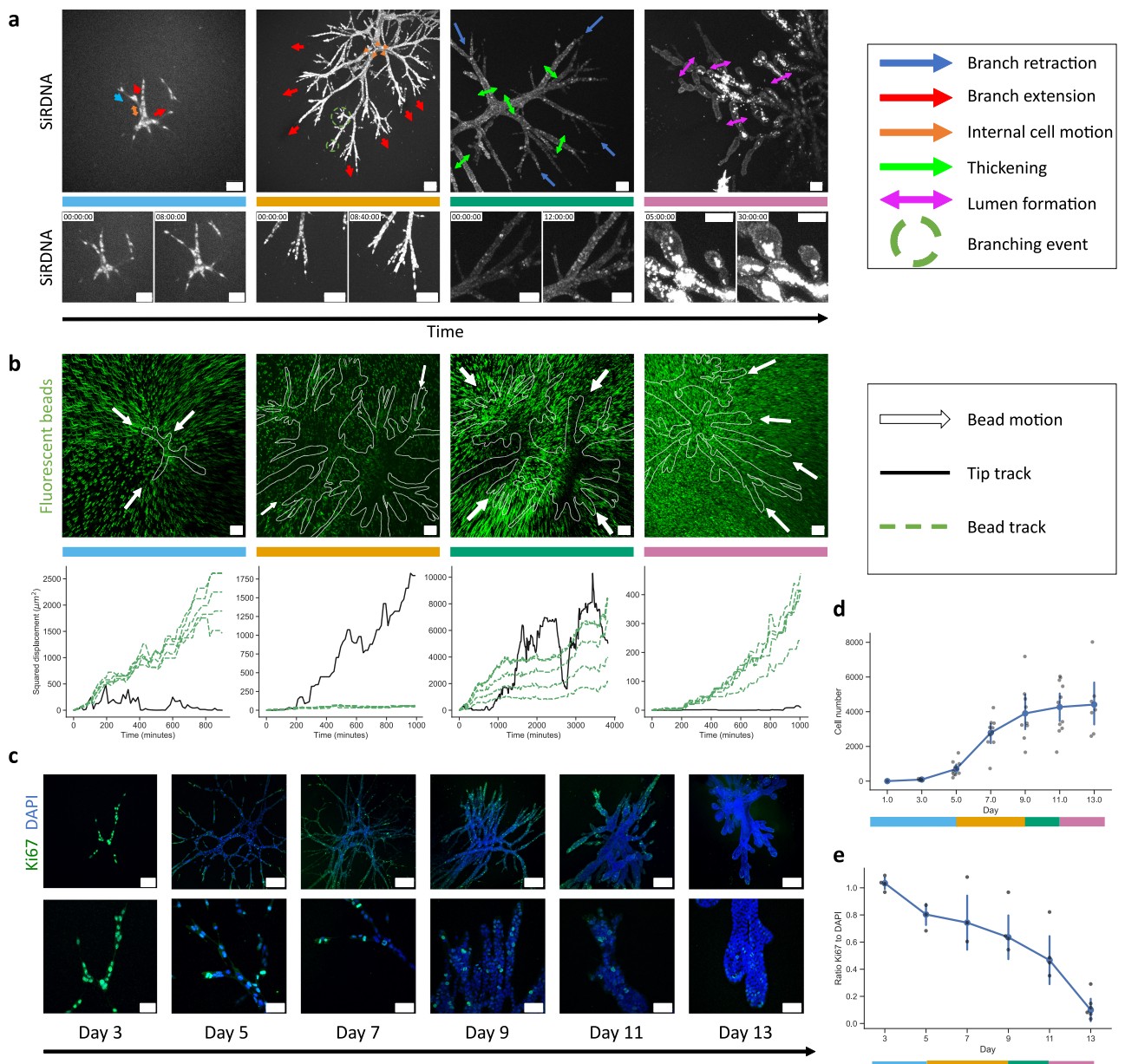

**Fig. 2 | Collagen-grown organoids go through different phases of development.** Development phases are denoted by color bars which follow the color code presented in Fig. 1. Blue: onset, orange: extension, green: thickening, pink: lumen formation. Organoids shown here are grown in collagen. **a** Cellular motion patterns observed with live confocal imaging for each development phase (*n* = 66 organoids). Cell nuclei are stained with SiRDNA (white). Scale bars: 100 μm. From left to right: Day 4 SUM projection, and Day 7, Day 10, Day 13 maximum projections. **b** Top, time-projections of fluorescent beads (green, maximum projections) trajectories at different time points, indicating the deformation field around the organoids. Organoids are outlined in white and white arrows denote the direction of bead motion. From left to right: Day 4–5, Day 7–8, Day 8–9, Day 13–14. Scale bars: 100 μm. Bottom, corresponding representative squared displacement of a branch tip (solid black) and the motion of beads (dashed green) in front of it, for each development phase. **c** Immunostainings of Ki67 (green) and DAPI staining (blue) in organoids at different time points. Top scale bars, from left to right: 80 μm first picture, 200 μm second picture, 200 μm for the rest. Bottom (zoom-in of the top row images) scale bars: 50 μm. Confocal slices. **d** Cell number evolution in organoids, estimated based on maximum projections of DAPI stainings (*n* = 56 organoids). Blue line indicates the mean tendency. Error bars: 95% confidence interval (CI). **e** Ratio of Ki67- over DAPI-positive cells (*n* = 24 organoids). Blue line indicates the median. Error bars: standard deviation.

$w \approx w_0(1 - k_b/k_d)$, and growth is still exponential but slower as limited by the branching rate $k_b$. Qualitatively, the key mechanism is that growth of a single branch via tip invasion is linear in time $v_0$, while proliferation is exponential in time. Thus, the former is always expected to be dominant at short time and the latter dominant at longer times. This contrasted with what the model would predict for 2D cell monolayers on a flat substrate, or 3D spherical growth in Matrigel relevant for unbranched organoids. In those cases, the edge's geometry limits the outer area to grow only linearly in time, whereas branching of tip cells allows for exponential growth of the effective "leading edge" in branched organoids, rapidly limiting the initial exponential growth phase of the volume, and resulting in a much faster decrease of proliferation (see SI Note for details).

Importantly, we found that this minimal model, fitted only on a live-short term time course between day 7 and 9 of organoid growth, could very well describe the entire time course of cell numbers from the Onset phase at day 1 to the early thickening phase at day 9 (Fig. 4b, d), where the main branched architecture of the organoids is

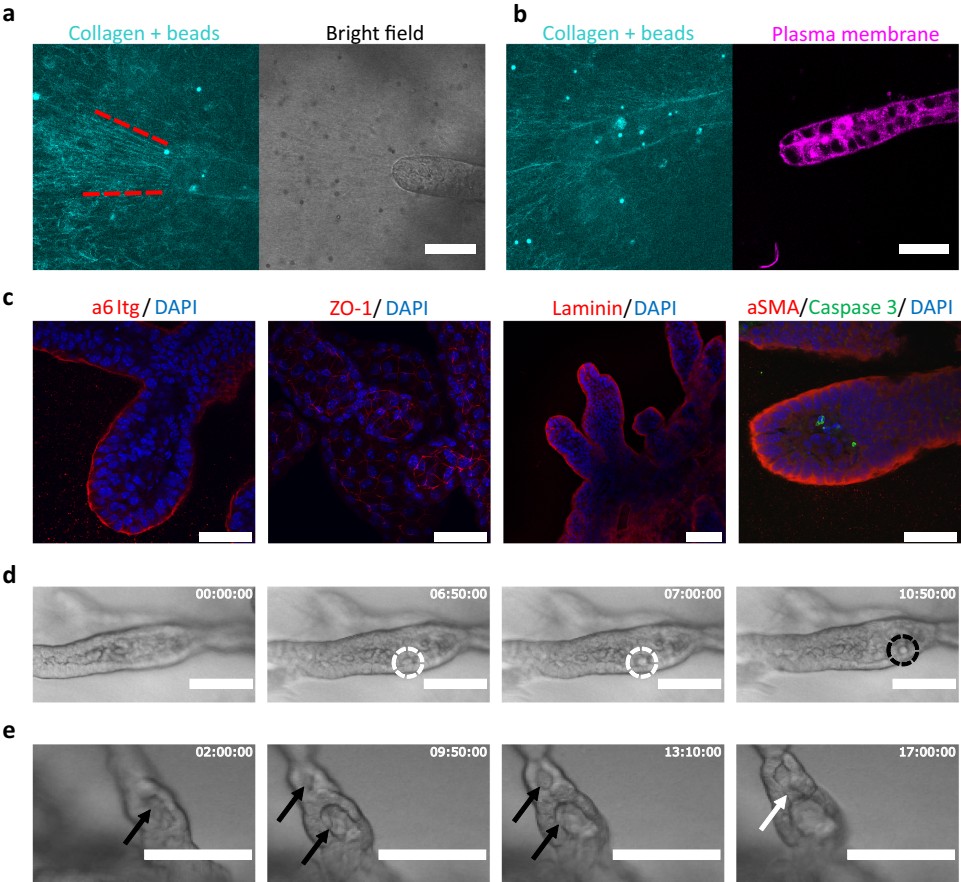

**Fig. 3 | Organoids remodel their surrounding ECM, and develop lumens via different processes.** All organoids shown here are grown in collagen. **a** Collagen fibers visualized with reflection microscopy (cyan), in front of a thickening branch at Day 10 ($n = 2$ replicates). The branch is pulling on the fibers in front of it, aligning them in a cone, outlined by the red-dashed lines. Scale bar: 50 μm. **b** Collagen fibers visualized with reflection microscopy (cyan) around the branch of an organoid at Day 10 (plasma membrane stained in magenta, $n = 2$ replicates). Scale bar: 50 μm. **c** From left to right: staining of DAPI (blue) and α6 integrins (red) a mediator of cell-ECM adhesion. Staining of DAPI (blue) and ZO-1 (red), a tight-junction associated protein. Staining of DAPI (blue) and Laminin (red) a major component of the basal lamina. Staining of α-smooth muscle actin (aSMA, red) a protein associated with contractility, Caspase 3 (green) denoting apoptotic cells, and DAPI (blue). Scale bars from left to right: 50 μm, 50 μm, 100 μm, 50 μm. All organoids Day 13, $n = 3$ individual experiments. **d** Time-lapse of an extrusion process at Day 11–12: dashed circles indicate cells initially in the epithelial-like layer lining the lumen, being extruded into the central cavities. White and black circles denote two different cells. Scale bars: 100 μm. **e** Time-lapse of microlumen nucleation and fusion at Day 12–13. The black arrows indicate existing microlumens undergoing swelling. The two microlumens end up resolving in a single lumen shown by the white arrow. Scale bars: 100 μm. Fluorescence images shown are all confocal slices except for the ZO1/DAPI and aSMA/Caspase 3/DAPI images in **c** which are maximum projections.

established and prior to lumen formation, which increases branch width due to different physical and osmotic forces. To make further predictions, in particular for the variability of branched organoid size and the dependency of branch width on not only time, but also the branch generation number, we also implemented a spatial version of the model[20,21], with the same ingredients and parameters as discussed above (Supplementary Note for details, and Fig. 4d–h, Fig. S6 for typical simulation outputs).

Firstly, we measured the average number of branches of organoids from day 1 to day 9, and found that it was well-fitted by exponential growth with a single rate $k_b$ invariant throughout time, as assumed in the model (Fig. 4f). Secondly, our model also predicted that the variability, as assessed by the standard deviation in branch number, across organoids should also grow exponentially—characteristic of a stochastic branching process. This described very well the data qualitatively (Fig. 4h), but also quantitatively (Fig. 4g) as a large fraction of the standard deviation, which we found to be on the same order of magnitude as the average itself, was explained by this simple model. Although this does not exclude other contributions to the variability across organoids, such contributions are expected to be additive, arguing that most of the heterogeneity in branching organoid

size could arise from the intrinsic stochasticity of branching, rather than intrinsically different average rates of branching.

Thirdly, our model predicted that the average branch width should initially grow during the first days of organoid growth, before plateauing in time to a near constant value (Fig. 4e), which was well-captured in the data when examining the average width of all branches. However, the spatial model also predicted a dependency of branch width on branch generation numbers: "terminal" branches (with an elongating tip) should plateau to smaller values of width, while non-terminal branches are not thinned-out by elongation and should plateau to larger values of width. Importantly, plotting separately terminal and non-terminal branch width revealed good qualitative and quantitative agreement with these predictions (Fig. 4e).

Finally, the model can be tested by observing the influence of batimastat treatment (a matrix metalloproteinase inhibitor, further described in the following section), which was found to abolish tip invasion speed ($v_0 \approx 0$). While branch volumetric growth is also perturbed in this condition (−77%), it remains active to a degree, so that by simple conservation law (elongation counteracts volumetric growth by promoting branch thinning in the model), we predict it to cause thickening of organoids (Fig. S6b, c, effectively to the maximum

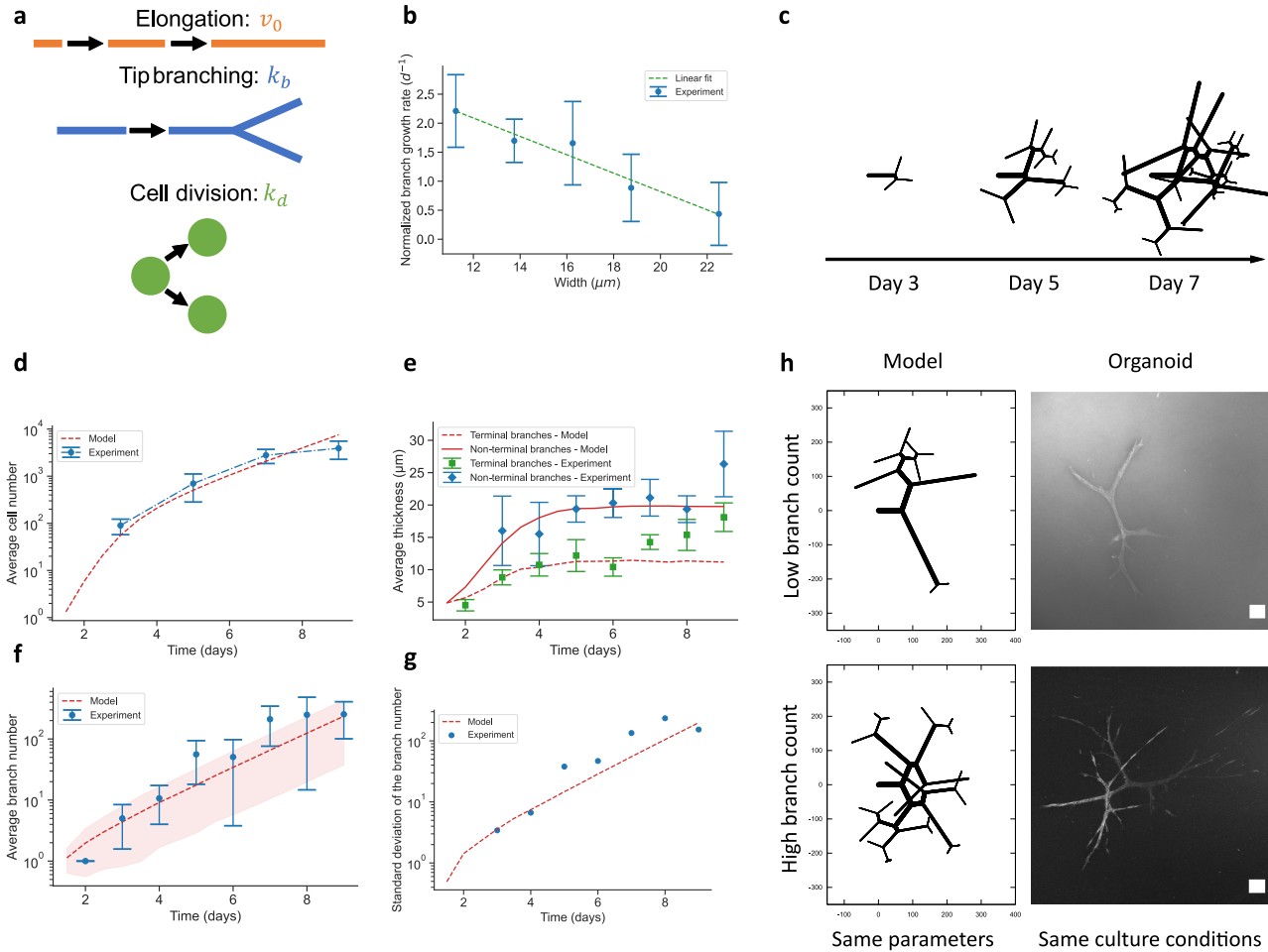

**Fig. 4 | A minimal biophysical model captures the main morphogenetic dynamics from the onset phase up to the thickening phase. a** Schematic representation of the processes considered in the model: branch elongation speed $v_O$, branching rate $k_b$, and proliferation rate $k_d$. **b** Evolution of the normalized volumetric growth rate of branches according to their width for PDAC organoids (blue, $n = 85$ points, $N = 3$ organoids). Data is extracted from live imaging organoids between day 7 and 10. We find that a negative linear dependency (green, dashed line) fits well the relationship between growth rate and branch width, and allows us to extract the maximal division rate $k_d$ and the maximal branch width at which proliferation is abrogated $w_0$. Error bars: mean ± standard error of the mean. **c** Spatial simulation of the branching process over time in pancreatic organoids using the determined PDAC organoids' growth parameters. **d** Evolution of the cell number over time for experimental PDAC organoid data (Experiment, blue solid line, mean ± sd, $n = 55$ organoids) and model predictions (red-dashed line).

**e** Evolution of the measured mean branch width over time for terminal branches ($n = 1420$ terminal branches, $N = 74$ organoids) and for non-terminal branches ($n = 123$ non-terminal branches, $N = 53$ organoids), and predictions of the spatial model. Error bars: standard error of the mean. **f** Evolution of the branch number per organoid over time ($N = 65$ organoids) and prediction of the model. The blue dots indicate the mean tendency. Error bars: standard deviation. **g** Evolution of the measured standard deviation of the number of branches per organoid over time and prediction of the model. **h** Comparison between organoid shapes simulated by the spatial model (left column) and actual organoids (right column, plasma membrane stain, summed slice projection), both at Day 5. Note that due to the stochasticity of the branching process, the simulated organoids can capture the phenotype diversity in the number of branches, even though the simulations parameters identical in the top and bottom panels. Scale bars: 100 μm.

---

thickness $w_0$ at long times for fully abolished tip invasion speed $v_0 \approx 0$), consistent with experimental observations (Fig. 5c, Fig. S6f), as described above. However, if proliferation is inhibited, as with an aphidicolin treatment (a proliferation inhibitor, further described in the following section), the model predicts thinning of the organoid in time (Fig. S6d), with nascent branches being particularly fragile, as because of their low cell number, linear tip migration is initially dominant compared to proliferative expansion, causing initial thickness decrease in the model (Fig. S6h, Movie S13) and thus potential breakage.

### Distinct dynamically regulated transcriptional programs orchestrate branching organoid development

To understand the underlying biological mechanisms of the developmental stages, we performed gene expression profiling at key stages of morphogenesis, the extension phase (day 7) and lumen formation

phase (day 13) (Fig. 5a). Morphologically, the extension phase is characterized by tip cell invasion and branch elongation. In this phase, cell proliferation signaling, indicated by Myc target signature as well as E2F signaling, is significantly enriched. In addition, increased basal-like and EMT signatures indicate mesenchymal de-differentiation associated with branch invasion in the extension phase. Furthermore, we observed highly enriched integrin and focal adhesion signaling, ECM interaction and Rho GTPase pathways during branch extension. In contrast, we observe upregulation of ion channel transport genes in the lumen formation phase. These findings indicate that single PDAC cells are able to self-organize into complex tubular structures termed Terminal End Bud Branched Organoids (TEBBO) by executing distinct developmental phases, which is concomitant with a change in gene expression profiles. Specifically, tumor cells activate an EMT program during branch elongation and switch to epithelial differentiation once matured. To functionalize the results obtained by RNA profiling, and

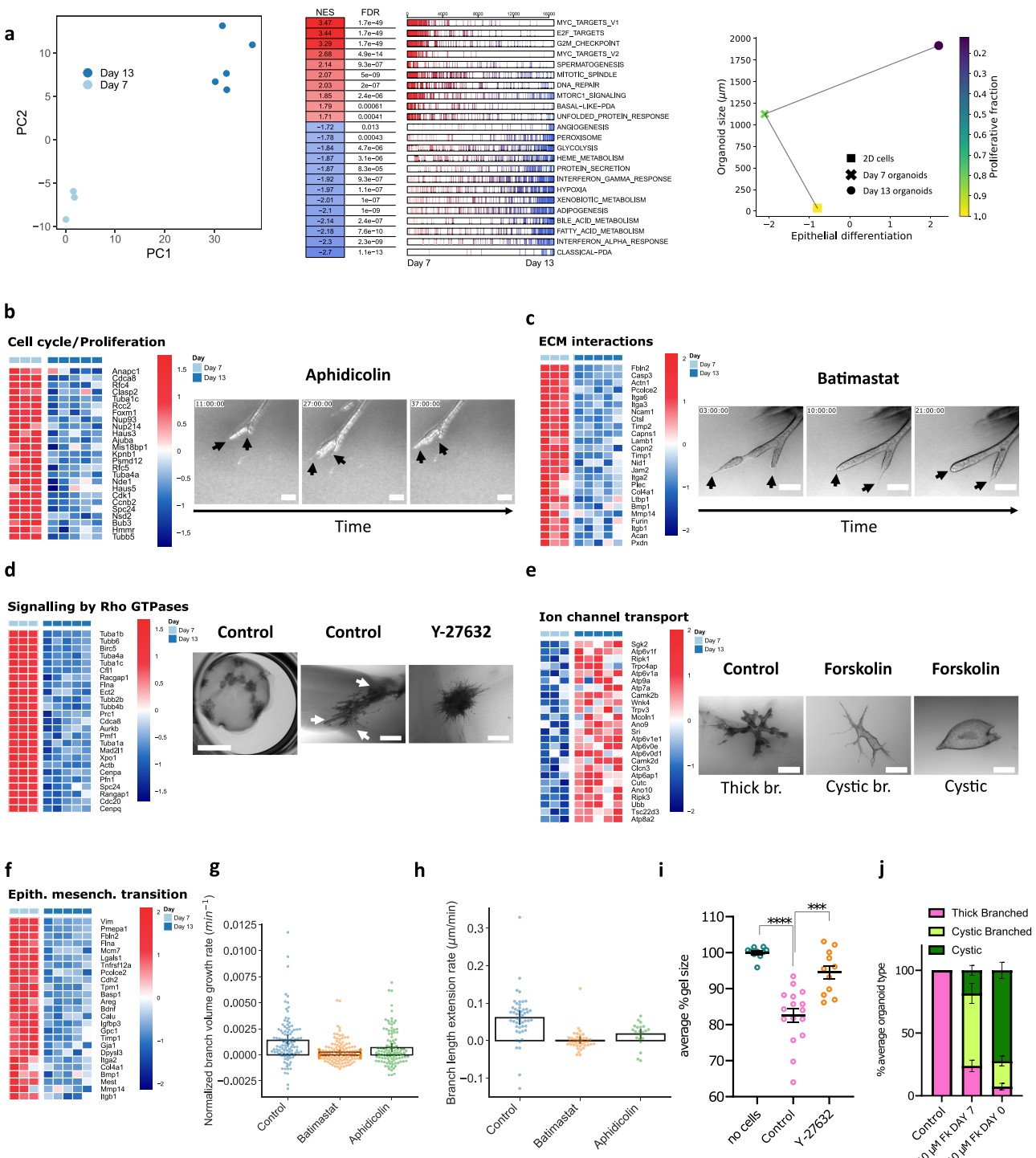

investigate the role of proliferation, matrix remodeling, contraction and ion channels in different phases, we manipulated these key processes using specific inhibitors.

**Organoid proliferation drives branch extension.** First, we imaged live organoids upon addition of aphidicolin, a proliferation inhibition drug at the Extension stage, day 7. We observed that after around 20 h, organoids lost their extensile phenotype, with branch tips rounding and retracting (Fig. 5b, Movie S13, Fig. S3a). Branches that were able to continue advancing into the matrix gradually became thinner at their centers before rupturing, indicating the importance of a sustained influx of cells to preserve branch integrity (Fig. S6h). Locally, we observed that when a branching event occurred after the addition of

aphidicolin, the newly formed branches were unable to extend equally, likely due to an insufficient amount of newly generated cells to fuel the growth (Fig. 5b, g, h). Finally, we observed that organoids maintained their integrity even 60 h after the addition of aphidicolin, and in spite of cell death events reminiscent of those happening at the thickening to lumen formation transition phase (Movie S13). These results indicate that while organoids are able to maintain an invasive phenotype for some time upon addition of aphidicolin, proliferation appears to be required to further fuel the extension of branches.

**ECM remodeling is required for branching.** Second, we investigated whether invasion via matrix metalloproteinases- (MMP) mediated degradation is crucial for in vitro branching morphogenesis, by

**Fig. 5 | PDAC organoids development is orchestrated by defined transcriptional programs. a**–**e** captions are given from left to right. Up- and downregulation genes heatmaps (log2 transformed) compare collagen-grown organoids at day 7 (D7) and 13 (D13). **a** Principal component analysis of bulk RNA sequencing of collagen-grown organoids at D7 (*n* = 3 individual experiments) and D13 (*n* = 5 individual experiments). Corresponding up- and downregulated clustered pathways. Developmental trajectory from 2D cells to D7 and D13 organoids. **b** Cell cycle- and proliferation-related genes heatmap. Time-lapse of an organoid branch upon addition of 2 μg.mL⁻¹ of aphidicolin at D7. Black arrows indicate spots of impaired branching. **c** Extracellular matrix- (ECM) related genes heatmap. Time-lapse of an organoid branch upon 10 μM batimastat addition at D8. Black arrows indicate stopped extension. **d** Rho GTPase signaling-related genes heatmap. Floating gel with D13 organoids in the well of a 24-well plate. Collagen surrounding an organoid at D13: white arrows indicate dense collagen regions due to organoid contraction. D13 organoid post-treatment with 5 μM Y-27632 added at seeding time. **e** Ion channel-related genes heatmap. D13 organoids in absence or presence of 10 μM forskolin added at seeding time. **f** Epithelial- and mesenchymal-related genes heatmap. **g** Normalized branch instantaneous volume growth rate for control (*n* = 103 points, *N* = 3 organoids), batimastat- (*n* = 142 points, *N* = 2 organoids), and aphidicolin-treated (*n* = 103 points, *N* = 3 organoids) organoids at the Extension stage. **h** Branch length extension rate for control (*n* = 51 branches, *N* = 3 organoids), batimastat- (*n* = 39 branches, *N* = 2 organoids), and aphidicolin-treated (*n* = 22 branches, *N* = 3 organoids) organoids at the Extension stage. **i** Ratio between the gel size at D13 and the gel size at seeding time, in absence of cells (*n* = 10 gels), with control organoids (*n* = 16 gels), and with Y-27632-treated organoids (*n* = 11 gels). Unpaired two-tailed parametric *t*-test with Welch's correction, two-tailed. ****$P < 0.0001$; ***$P = 0.0002$. **j** Organoid phenotypes distribution at D13, according to the addition day of 10 μM forskolin. (Control: *n* = 159, Fk from D0: *n* = 73, Fk from D7: *n* = 101 organoids). Scale bars in **b**, **c**: 100 μm; in **d**: 5 mm, 500 μm, 500 μm; in **e**: 500 μm Black bar plots in **g**, **h**: mean ± 95% CI; in **i**, **j**: mean ± sem.

applying two broad spectrum MMP inhibitors, namely marimastat and batimastat (marimastat 10 μM, batimastat 1 and 10 μM). Matrix metalloproteinases are well-characterized mediators of cancer cell invasion and metastasis, through the remodeling of the tumor microenvironment[22]. Out of the 28 known MMPs the most important for local degradation of the ECM in PDAC are MMP-2, −7, −9 & the membrane type −1 (MT1-MMP)[23], with various mechanisms of action including active invasion, cleavage of pro-MMP forms, migration, and cross-talk signaling. The most profound effect on the organoid phenotype was observed by using batimastat (10 μM) leading to more scattered and small organoids, as well as thin-branched organoids in comparison to thick-branched controls (Fig. S4). The fact that batimastat exceeded the effects of marimastat indicates the importance MT1-MMP in PDAC organoid invasion, which is also supported by increased MT1-MMP (MMP14) expression during day 7 (Fig. 5f and Fig. S4). We next aimed to define the most critical time points of MMP-driven branching. We administered batimastat either at day 0, day 3, day 5 (day 0–5, the onset phase), day 7 (day 6-8, the extension phase), day 9 (the thickening phase), and day 11 (the lumen formation phase). Time course treatments of the organoids indicated that MMP-activation at the onset and extension phase are most critical for the establishment of mature, thick-branched organoids (Fig. S4b, c). In addition, measuring the major axis length of time-course treated organoids revealed that the strongest effect on organoid size was observed when organoids were treated during the onset/extension phase (Fig. S4d). Live-cell imaging under MMP inhibition (batimastat 10 μM) at the extension phase revealed that leading cells retracted their invasive protrusions and branch tips started rounding up and thickening, thereby stopping the organoid extension (Fig. 5c, g, h, Movie S12). The results of the batimastat experiments, together with the aphidicolin inhibition results, highlight the combined role of both the invasive and proliferative processes in establishing thick-branched organoids, indicative of invasive proliferation as a driver of branching morphogenesis in our system.

**Contraction is required for branched tubular structure formation.** Third, in order to demonstrate the importance of the cytoskeletal activation and contraction via Rho-GTPase pathways as a mechanism of branching morphogenesis, we inhibited MYOII by administering 5 μM of Y-27632 (Rock inhibitor) from day 0. During organoid development, we observed a pronounced decrease in gel size suggesting tumor cell contractility. When measuring the size of the gels we saw a statistically significant relative decrease in gel diameter of 19%. Indeed, we were able to prevent gel size shrinkage almost to the full extent compared to control cells by the addition of Y-27632 (96%) (Fig. 5d, i). In addition to the reduced gel contraction, treated organoids failed to establish branched tubular structures (Fig. 5d, i). These data indicate that activation of the Rock pathway is required for appropriate organoid morphogenesis.

**Ion flux drives lumen swelling.** Fourth, to demonstrate the importance of ion flux in the lumen formation process we administered forskolin (10 μM) at distinct stages of organoid development, which induced lumen formation and swelling. As previously demonstrated, forskolin induces organoid-swelling by Cl-/Na+ influx due to cAMP-PKA pathway activation and CFTR phosphorylation[24,25]. When treating organoids from day 0 onwards, the organoids are forced to form large, mostly symmetric cysts (72%) or branched cysts (23%), whereas controls assume a thick branched morphology (Fig. 5e, j, Fig. S3b). Organoids exposed to forskolin from developmental day 7 favor the cystic branched phenotype with few symmetric cysts (Fig. 5e, j, Fig. S3b). These results highlight the importance of a regulated swelling, in time and in intensity, for the formation of thick-branched organoids connected by a seamless lumen.

We find that processes of proliferation, invasion, branching, contraction, and ion flux regulation, all play a role in the development of pancreas-derived branched three-dimensional structures connected by a single seamless lumen, and possessing terminal end buds.

We report that those processes display varying spatiotemporal dynamics in the course of development and highlight that the balance between those processes and their activity at the appropriate time-points and with appropriate intensities, as illustrated by perturbation experiments conducted with inhibitors at various timepoints, seems to be crucial for the proper establishment of Terminal End Bud Branched Organoids structures.

## Discussion

The ability to recover in vitro the tumor morphogenesis of complex tubular structures reminiscent of PDAC in vivo, opens the possibility to investigate the dynamics of tumor growth with high spatiotemporal resolution and precision. Organoids derived from human or murine pancreatic cancers embedded into Matrigel display only a uniform and symmetric spherical structure[1,3], preventing recapitulation of organ- or disease-like structure and function. Here, we demonstrate that PDAC cells, when embedded in collagen gels (type I) and brought to floatation, are able to develop into complex branched structures exhibiting tubular morphology and glandular differentiation, key morphogenetic features of PDAC[26]. This in vitro reconstitution of PDAC morphogenesis allows the direct and detailed observation of the dynamics of tubulogenesis, which leads to the classification of the developmental trajectory into four main phases, each with their own hallmarks, in terms of cell motion, physical interactions with the ECM, and proliferation patterns. The cell-of-origin of the described PDAC branched organoids show epithelial differentiation, as they are derived from epithelial clusters of *Kras*^*G12D*-driven pancreatic cancer[8]. The transcriptional program at distinct phases of organoid development indicates that the tumor cell differentiation dynamically correlates with the tubulogenesis. Specifically, we see a basal-like differentiation in the

extension phase and a classical gene signature in the mature lumen phase. This has important implications for drug testing and clinical translation, as transcriptomic subtypes define therapeutic vulnerabilities[27].

Our study joins a growing body of literature, investigating branched, more architecturally-faithful, organoids derived from various organs such as the lungs[28], the kidney[29], or the mammary gland[9,10], and represents an important step forward in the emerging field of three-dimensional pancreatic organoids[30,31].

In recent years, several studies have been aiming to connect the by-now well-characterized genetic events occurring in branching morphogenesis, to the actual physical fundamental processes that shape the structures. Some of the fundamental processes that drive branching morphogenesis in pancreatic organoids have been reported in other systems. For instance, tip branching plays a major role in the establishment of the kidney and the lung architecture[19], and ECM degradation has been shown to be critical in mammary gland organoids growth[10].

While each organ might possess a specific set of mechanisms for branching morphogenesis, both at the molecular and at the physical level[19,32], minimal biophysical models using simple local rules have been shown to recapitulate important structural properties such as size or network topology, across a range of organs, established through self-organization[20,33].

Here, our presented predictive analytical model gives insights into the fundamental mechanisms of tumor organoid growth, where it is the competition between proliferation and outgrowth that determines tumor organoid structure formation in the Onset and the Extension phases. The resulting structure encodes the signature of the growth parameters.

Our study opens perspectives where a manipulation of these model parameters should yield changes relatable in experimental morphology, thus providing a starting point to rationalize the relation between gene regulation, growth parameters, and the resulting self-organization of the structure.

## Methods

### Ethics declaration
For the endogenous mouse model[8] as well as the orthotopic transplantation model, mice were euthanized in compliance with the European guidelines for the care and use of laboratory animals. In detail, animals were euthanized when a palpable abdominal mass above 1.5 cm, ascites, signs of sickness, or a weight loss of >15% of body weight were detected. Mice were monitored on a daily basis regarding general health status as well as body weight, and housed under specific-pathogen-free conditions. Animal studies were approved by the Institutional Animal Care and Use Committees of Technische Universität München (Regierung von Oberbayern, Munich, Germany).

### Mouse background
For the endogenous mouse model[8], mice were maintained on *C57Bl/6;129S6/SvEv* mixed background, and female and male mice were randomly submitted to respective tumor cohorts. For the generation of double-mutants, pancreas-specific Cre lines were intercrossed with *KrasG12D-Panc* (PK mice). For the orthotopic transplantation, female athymic nude mice, aged between 7 and 9 weeks, with NU(NCr)-Foxn1nu background (Charles River) were used.

### Two-dimensional (2D) cell culture of PDAC cells
Primary tumor cells were collected from genetically engineered mouse models of pancreatic cancer: *Ptf1a$^{Cre/+}$;Kras$^{G12D/+}$* (KC mice)[8] or *Pdx1$^{Cre/+}$; Kras$^{G12D/+}$; TP53* ΔHO (KPC mice). For 2D cultures, cells were seeded in 75 cm² flasks in Dulbecco's Modified Eagle's Medium (DMEM)−high glucose supplemented with 10% v/v Fetal Bovine Serum (FBS) and 1x Penicillin/Streptomycin (all from Thermo Fisher Scientific), hereafter described as the "cell culture medium" or "PDAC medium", as described in a previously published study[8].

Cell culture medium was fully exchanged every 2–3 days. Upon confluence, cells were passaged using 0.05% Trypsin-EDTA (Thermo Fischer Scientific). Cells were cultured in an incubator with a humidified atmosphere supplemented with 5% $CO_2$ at 37 °C. The procedure is schematized in Fig. S7a.

### Two-dimensional (2D) cell culture of pancreatic ductal cells (PDCs)
The healthy adult pancreatic ductal cells (PDCs) were cultured as previously described[34]. Briefly, cells were seeded on collagen coated plates (a 3 mL collagen type I layer (2.31 mg/mL) on a tissue culture dish), and were grown in PDC medium: DMEM/F-12 (Thermo Fisher Scientific), 5 mg/mL D-glucose (Sigma Aldrich), 0.5% ITS premix (Corning), 5% Nu-Serum (Corning), 1x Penicillin/Streptomycin (Thermo Fisher Scientific), 25 µg/mL Bovine Pituitary Extract (Thermo Fisher Scientific), 100 ng/mL Cholera Toxin (Sigma Aldrich), 1 µM Dexamethasone (Sigma Aldrich), 10 mM Nicotinamide (Sigma Aldrich), 100 µg/mL Primocin (Invivogen) and 20 ng EGF (R & D systems) (Table S1). Media changes were performed every 48 h and upon 80–85% confluency the collagen was further digested for 15 min at 37 °C with 1.5 mg/mL Collagenase Type 4 (Worthington) diluted in DMEM/F-12, then cold PBS was added and the mixture was centrifuged. The cell pellet was then trypsinized and Soybean Trypsin Inhibitor (STI) was used to quench the effect of trypsin. Afterwards, 10.000 cells were seeded into collagen gels (as described in "Organoid preparation"). The procedure is schematized in Fig. S7b.

### Organoid preparation
For collagen-grown organoids, we adapted a previously described protocol[9]. 2D cells were detached using trypsin, and a series of dilutions was prepared in order to reach a final concentration of 500 cells/mL media. Then, the following components were added: cell culture medium, cell suspension, neutralizing solution (550 mM HEPES in 11xPBS) and Collagen Type I (rat tail from Corning), mixed gently and incubated for 1 h at 37 °C in order to polymerize. Afterwards cell culture media was added and the gels were loosed up (detached) with the help of a 0.1–10 µL tip. The final collagen concentration used is 1.3 mg/mL.

Media was changed first after 72 h and then every 48 h, growth factors and inhibitors were renewed according to media changes unless stated otherwise. For live-cell imaging, unless mentioned otherwise, drugs were added immediately prior setting up samples for imaging. In the case of Matrigel-grown organoids, resuspended cells were mixed with growth factor-reduced Matrigel (Corning) and left to polymerize for 1 h in 37 °C and seed in domes or gels. For the mixture of collagen gels with Matrigel, first all components of the collagen gels were added and then Matrigel.

### Orthotopic implantation into mice
Mice were anesthetized using MMF (5 mg/kg midazolam, 500 µg/kg medetomidine, 50 µg/kg fentanyl) and after a small abdominal incision the spleen was exposed by gentle pull. 2500 cells were carefully injected into the pancreas using a microliter syringe with a 27-gauge needle. Thereafter, the incision was closed and MMF anesthesia was antagonized by injecting AFN (750 µg/kg atipamezole, 500 µg/kg flumazenil, 1.2 mg/kg naloxone). Mice were monitored postoperatively on a daily basis regarding general health status as well as body weight. After 2–3 weeks, mice were sacrificed, and tumor tissue was harvested and fixed with 4% PFA.

### Immunostaining
The organoids were carefully washed and fixed using 4% paraformaldehyde (Alfa Aesar) for 15 min at room temperature. For

immunofluorescence imaging, cells were permeabilized with 0.2% Triton-X 100 (Sigma) in DPBS (Gibco) for 10 min at RT, blocked overnight at 4 °C with 10% normal donkey serum/0.1% BSA (Carl Roth) in DPBS, then labeled with primary antibodies diluted in 0.1% BSA/DPBS overnight at 4 °C. Secondary antibodies were incubated for 3 h at room temperature and DAPI was used for staining the nuclei. All antibodies used are listed in Supplementary Materials Tables 2 and 3.

Immunostaining images were acquired using a laser scanning confocal microscope (Olympus FluoView 1200; Olympus Corporation) equipped with an Olympus UPlanSAPO ×60 1.35, UPlanSAPO ×40 1.25 solid immersion lens oil immersion objectives and UPlanSAPO ×20 × 0.75, UPlanSAPO X10 0.40 air immersion objectives (Olympus).

## Histology
Organoids were fixed as described above in 4% PFA. For orthotopically implanted tumors, tissues were fixed in formalin (10%) overnight, dehydrated, and embedded in paraffin. Hematoxylin and eosin (H&E) staining was performed as previously published[1]. Briefly, paraffin-embedded sections were dewaxed in xylene (two times, 5 min each), and rehydrated first in isopropanol (2 times, 5 min each), and then in decreasing ethanol concentrations (at 96% two times, 2 min each, and at 70% two times, 2 min each). Sections were rinsed with distilled water for 25 s, and were stained with Mayer's Hematoxylin for 8 min. Sections were then rinsed in tap water for 10 min, before applying a 1% alcoholic solution of eosin for counterstaining, for 4 min. Following this, the slides were passed in ethanol (96%, 30 s), isopropanol (2 times, 25 s each), and xylene (2 times, 1 min 30 s each).

For the immunofluorescence of tissue and organoid sections, slides where first deparaffinized, then immersed into unmasking solution (Vector Laboratories) for 10 min at 360 V in a microwave, and afterwards washed sequentially with dH₂O and DPBS and blocked (0.5% BSA/0.5% Triton-X 100 in DPBS) for 1 h at room temperature. Primary antibody diluted in blocking solution was added for overnight incubation. Next, slides were washed 3x with DPBS and secondary antibodies were incubated for 1 h at room temperature and DAPI was used for staining the nuclei.

## Tissue clearing
Organoids were incubated overnight at 4 °C in the dark with FUnGI solution as previously published[35], in order to obtain a fully transparent collagen gel, and then mounted on a 2-well Ibidi slide for confocal microscopy.

## RNA-isolation
Cells grown in 2D cultures were directly collected in RLT buffer with β-Mercaptoethanol, while the 3D organoids were first digested for 12–15 min at 37 °C with 1.5 mg/mL Collagenase Type 4 (Worthington) in DMEM supplemented only with Penicillin/Streptomycin until complete matrix digestion. The organoids then were washed once with DPBS and further collected in RLT buffer with β-Mercaptoethanol until further use. Before the RNA isolation, we homogenized the cells/organoids lysates using QIAshredder (Qiagen). Total RNA was isolated using the RNeasy Plus Micro Kit (Qiagen) according to the manufacturer's instructions, with the addition of a 15 min on column DNA digestion step using RNase-Free DNase set (Qiagen).

## RNA-sequencing
Library preparation for bulk-sequencing of poly(A)-RNA was done as described previously[36]. Briefly, barcoded cDNA of each sample was generated with a Maxima RT polymerase (Thermo Fisher) using oligo-dT primer containing barcodes, unique molecular identifiers (UMIs) and an adapter. Ends of the cDNAs were extended by a template switch oligo (TSO) and full-length cDNA was amplified with primers binding to the TSO-site and the adapter. NEB UltraII FS kit was used to fragment cDNA. After end repair and A-tailing, a TruSeq adapter was ligated and 3'-end-fragments were finally amplified using primers with Illumina P5 and P7 overhangs. In comparison to Parekh et al.[36], the P5 and P7 sites were exchanged to allow sequencing of the cDNA in read1 and barcodes and UMIs in read2 to achieve a better cluster recognition. The library was sequenced on a NextSeq 500 (Illumina) with 63 cycles for the cDNA in read1 and 16 cycles for the barcodes and UMIs in read2. Data was processed using the published Drop-seq pipeline (v1.0) to generate sample- and gene-wise UMI tables. Reference genome (GRCm38) was used for alignment[37]. Transcript and gene definitions were used according to the GENCODE Version M25. Heatmaps shown display the log2 fold change.

## Statistical analysis of gene expression data
High-throughput gene expression data from the conditions indicated in the text were carried out using the R environment for statistical computing[38] (v4.0.4).

## Differential gene expression analysis
Genome-wide differential gene expression analysis for RNA-Seq count data was carried out using a negative binomial generalized linear model as implemented in the DESeq2 R package[39] to test for differentially expressed genes between experimental conditions. For dispersion estimates we considered the following covariates: cell line, genotype, dimension (2D, 3D), extracellular matrix composition (none, collagen, Matrigel) and time (7 days, 13 days). For individual comparisons, a false discovery rate (FDR) < 0.1 was considered significant.

## Gene set enrichment analysis
Gene set enrichment analysis (GSEA) was carried out on individual differential gene expression signatures between two conditions as represented using the *fgsea* R package[40] and using Wald statistics as gene-level statistics. Gene sets were retrieved from the MSigDb v7.3[41,42]. Enrichment results for select pathways were illustrated using custom R code. For select pathways, leading edge genes were illustrated between two conditions after scaling all rows to have mean 0 and variance 1 (Z-score transformation) using the *pheatmap* R package[43].

## Extreme limiting dilution analysis
Extreme Limiting Dilution Analyses (ELDA) were performed according to previously described protocols[44]. Briefly, after a series of the following dilutions: 50.000 cells/mL, 2.500 cells/mL and 500 cells/mL, cells were seeded into floating collagen gels at an extremely low density (sub-clonal of 0.75 cells/gel) and left to form organoids for 13 days. Then, the number of positive reactions (gel containing at least one organoid) were measured for the primary structures and then passaged into 0.75 cells/gels to form secondary structures. The same analysis was performed and repeated for the tertiary structures. Using the Walter and Eliza Hall Institute website[45], we calculated the potency of cells to form multicellular structures and plotted the logarithmic fraction of non-responding gels to cell dose.

## Chemical perturbations
To perturb mitosis, aphidicolin (Sigma A4487) was used at 2 µg.mL⁻¹ (concentration in medium) for live imaging. To inhibit matrix metalloproteinases (MMP) activity, marimastat (Sigma M2699) at 10 µM, and batimastat (Sigma SML0041) at 1 µM and 10 µM were used in drug screening.

For live-cell imaging, we used batimastat at 10 µM (concentration in medium).

To induce organoid swelling by Cl-/Na+ influx, we used Forskolin (Sigma F6886) at 10 µM. For live imaging, unless mentioned otherwise, drugs were added immediately prior setting up samples for imaging. To act on the Rho-GTPase pathways and inhibit MYOII, we used the Y-27632 Rock inhibitor (Biomol 10005583) at 5 µM.

## Live-cell imaging

Live imaging was performed using a Leica TCS SP5 II confocal microscope (software LAS AF version 2.6.3.8173) and a Leica DMi8 confocal microscope (software LAS X version 3.5.5.19976). Live-imaged samples were kept at 37 °C in a 5% $CO_2$ atmosphere using an ibidi Stage Top Incubation System (ibidi 10722).

For live imaging, cell nuclei were labeled using SiRDNA (Spirochrome SPY650-DNA SC501) for a minimum of 3 h before measurement at 1–2 µg.mL$^{-1}$ (concentration in collagen + medium volume).

To avoid potential interference due to phenol red when imaging with SiRDNA, we used Dulbecco's Modified Eagle's Medium−high glucose no phenol red (Thermofisher 21063-029) + 1:10 Fetal Bovine Serum (Sigma F0804), hereafter described as the "observation medium".

Plasma membranes were labeled using CellMask Deep Red (Thermofisher C10046) at 0.1%.

## Image acquisition

The LAS AF (version 2.6.3.8173, Leica) and LAS X (version 3.5.5.19976, Leica) softwares were used for live-cell imaging and for reflection imaging. Immunostaining images were acquired using the Olympus FluoView 1200 software (Olympus Corporation). RNA library were sequenced on a NextSeq 500 (Illumina).

"Gel overview" images shown in Fig. S2c were acquired using the Tile Scan function of a Leica DMi8 Thunder microscope, and stitched using the "Mosaic merge" function.

## Image and data analysis

Images were analyzed using ImageJ 1.53c[46] or ilastik (ver 1.3.3)[47]. Three-dimensional image reconstruction was performed using Imaris (8.2.0, Oxford Instruments).

Numerical data were analyzed using Python (ver 3.7.6) with tools from the SciPy package (ver 1.6.2), Graphpad Prism (ver 9.0.2) and Wolfram Mathematica 10. Graphs were produced using Python or Graphpad. Figures were assembled using Inkscape (ver 1.0beta2). BioRender.com was used to produce Supplementary Fig. 2c', 7a, b.

## Organoid major axis estimation

The sizes of control collagen-grown and Matrigel-grown organoid were determined using ilastik Object Classification routine to segment bright field images and extract the major-axis of an automatically fitted ellipse.

As long-term batimastat-treated organoids tend to show more fragmented phenotypes, less-well detected by ilastik, their major axis was measured by manually fitting an ellipse using ImageJ.

## Bead branch tracking

To monitor the deformation field generated by organoids in the ECM, fluorescent beads (Fluoresbrite YG Microspheres 3.00 µm, Polysciences 17155-2) were added at cell seeding time in the non-polymerized collagen mixtures.

We performed a manual tracking of branch tips and of fluorescent beads positions using ImageJ "Manual tracking" plugin on confocal live imaging full stacks or with maximum projections when the organoid growth remains in focus. Beads are chosen in a cone in front of active branch tips.

## Ki67 to DAPI ratio estimate

Immunostaining pictures of organoids stained against DAPI and Ki67 at 10x magnification were used to estimate the ratio of Ki67-positive to DAPI-positive cells over time. The colored jpg images of each channel were converted to grayscale using ImageJ. We then used the Cell Density Counting routine of ilastik to analyze each channel dataset separately.

A subset of cells was labeled before running the algorithm and manual corrections were performed as needed to refine the detection. As the ilastik Cell Density Counting routine tends to overestimate the cell numbers at early time points for DAPI and Ki67, and at late time points for Ki67, we also performed a fully manual counting to correct the data points showing aberrant orders of magnitude, using ImageJ Cell Counter. Ratios are computed for each organoid by dividing the number of counted Ki67-positive cells by the number of detected cells in the DAPI channel.

To assess the effect of batimastat and Y-27632 on the proliferative capabilities of organoids, organoids were treated continually, starting at seeding time with either of the drug, fixed at day 7 or 13, stained against DAPI and Ki67, and compared to non-treated organoids also stained against DAPI and Ki67 at the same timepoints. To estimate the number of DAPI- and Ki67-positive cells, we used the Cell Density Counting routine of ilastik. The channels were processed separately. In addition, the day 7 and day 13 images were processed separately.

## Cell nuclei number determination

Cell numbers were estimated using maximum projections images of DAPI-stained organoids.

For the semi-automatic counting, projections were loaded in ilastik and processed using the Cell Density counting routine: a subset of the nuclei data was manually labeled before running the density estimation algorithm. Iterations of manual corrections were performed as needed to refine the detection.

We also performed a fully manual counting of a subset of the same maximum projections, up to day 9, using the Cell Counter plugin of ImageJ to ascertain the order of magnitude given by the ilastik algorithm.

## Branch thickness measurement (live)

For the dynamic branch thickness measurements, we used ImageJ line drawing tool on bright field channel movies acquired with confocal microscopy. We measured the tip width by drawing a line 30 µm behind the branch tip. When the branch possesses a spiky phenotype, we ignore the leading protrusion and measure the width 30 µm behind the beginning of the protrusion.

The "body" measurements are measured using the same protocol, but at a distance of 100 µm.

## Branch thickness measurement (static)

For static branch thickness measurement, we used ImageJ line drawing tool on summed projections images of organoids stained with CellMask at 0.05%, acquired with confocal microscopy. We measured the terminal branch width by drawing a line 30 µm behind the branch tip. When the branch possessed a spiky invasive protrusion at the tip, we ignored the leading protrusion and measured the width 30 µm behind the beginning of the protrusion. During the Onset phase, to account for nascent terminal branches that are <30 µm long, we measure the width at the base of the terminal branch, close to the branching point.

The "body" measurements are measured using the same protocol, but at a distance of 100 µm of the terminal branch tip. We also measure the width of non-terminal branches "after a branching point", at a distance of 100 µm after the base of the "Y" shape defined by two branches meeting each other.

## Branch length measurement (live)

For the dynamic branch length measurements, we used ImageJ line drawing tool on the bright field channel acquired with confocal microscopy. We measure the branch length by drawing a segmented line starting from the tip of a branch, including the leading spiky protrusion if there is one, and going up to the nearest branching point. When plotting all branch lengths on the same graph, we normalized the time by setting zero as the first frame in which a branch is tracked.

### Branch counting (static)

Organoids at different timepoints were fixed and stained with Cell-Mask at 0.05% to label their plasma membranes. Confocal microscopy was used to acquire z-stacks, which were then reconstructed in three dimensions and analyzed on Imaris. A "branch" was defined as the segment between a "tip" and a branching point (in this case a so-called "terminal branch"), or between two branching points (in this case a so-called "non-terminal branch"), and manually labeled.

### Branching event counting

Branching events were manually counted on live imaging confocal stacks using ImageJ. We consider the leading 6 cells of a branch. True-Yes: indicates that a branching event was preceded by a proliferation event. True-No: indicates that a branching event occurred without being preceded by a proliferation event. False-Yes: indicates that a proliferation event occurred but was not followed by a branching event.

### Collagen reflection microscopy

Collagen fibers were imaged using the Reflection mode of a SP5 II confocal microscope and of a Leica DMi8 confocal microscope, with the 488 nm laser line.

To visualize the collagen architecture surrounding the organoids while avoiding auto-fluorescence artefacts from the cell membranes when using reflection microscopy, we incubated live organoids with Triton-X 100 diluted at 10% in PBS for a minimum of 1 h in order to degrade the cell membrane.

Organoids were then washed one time with PBS, fixed for 15 min in 4% PFA at room temperature, and washed again four times with PBS for a total of 20 min. Organoids were either stored at 4 °C in PBS or imaged immediately.

We used CellMask (Thermofisher C10046), a plasma membrane staining dye, at 0.1%, to ensure that the cell membranes were properly degraded.

### Linear regression fits

Linear regression fits were performed using the Python Stats Linregress function.

### Local thickness analysis

Local thickness heatmaps were produced using ImageJ Local Thickness (complete process) plugin[48] on maximum projection of confocal stacks.

### Statistical analysis

Graphpad Prism (ver 9.0.2) and the R environment for statistical computing (v4.0.4) were used for statistical analysis. No particular statistical method was used to define sample size and no specific hypothesis was tested.

For chemical perturbations and immunostaining experiments, a minimum of three independent experiments were performed.

For live imaging morphometrics characterization, 2–4 replicates were measured for each set of experiment, and experiments were performed at least twice with similar outcomes.

95% Confidence intervals are computed via bootstrapping with 1000 bootstrap iterations on Python using the *seaborn* package.

### Reporting summary

Further information on research design is available in the Nature Research Reporting Summary linked to this article.

## Data availability

Source data used to generate the graphs in this manuscript are available on a Zenodo repository with the identifier https://doi.org/10.5281/zenodo.6577226. The RNA sequencing data that support the findings of this study are deposited in the GEO database under accession code GSE200308. Additional data, such as confocal stacks, that support the findings of this study are available from the corresponding authors upon request. Source data are provided with this paper.

## Code availability

Source code is available on a Zenodo repository with the identifier https://doi.org/10.5281/zenodo.6577226[49].

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

## Acknowledgements

A.R.B. acknowledges the financial support of the European Research Council (ERC) through the funding of the grant Principles of Integrin Mechanics and Adhesion (PoINT) and the German Research Foundation (DFG, SFB 1032, project ID 201269156). E.H. was supported by the European Union (European Research Council Starting Grant 851288). D.S., M.R., and R.R. acknowledge the support by the German Research Foundation (DFG, SFB1321 Modeling and Targeting Pancreatic Cancer, Project S01, project ID 329628492). C.S. and M.R. acknowledge the support by the German Research Foundation (DFG, SFB1321 Modeling and Targeting Pancreatic Cancer, Project 12, project ID 329628492). M.R. was supported by the German Research Foundation (DFG RE 3723/4-1). A.P. and M.R. were supported by the German Cancer Aid (Max-Eder Program 111273 and 70114328).

## Author contributions

A.P., A.R.B., C.S., M.R. and S.R. conceived the project and the experiments. A.P., M.S., S.R., G.Z. performed the experiments and analysis. D.S. characterized and provided primary cell lines. K.P. and M.R. performed animal experiments. K.S. performed histological analyses. A.P. isolated the RNA, R.Ö. and R.R. performed RNA seq and H.C.M. analyzed the data thereof. The theoretical model was developed by E.H. and validated together with S.R., G.Z., and A.R.B. A.P., A.R.B., E.H., M.R., and S.R. wrote the paper. All authors reviewed and approved the manuscript. These authors contributed equally: A.P., S.R. These authors jointly supervised this work: A.R.B., M.R.

## Funding

## Competing interests

The authors declare no competing interests.
