## [Peer review file · Nature Communications]

REVIEWER COMMENTS

Reviewer #1 (Remarks to the Author): Expert in pancreatic cancer organoids and genomics

In the manuscript entitled “Spatiotemporal dynamics of self-organized branching in pancreas-derived organoids” the authors establish in vitro culture conditions that enable culturing of branching murine PDAC (pancreatic ductal adenocarcinoma) organoids. They use this system to show that organoids established from KrasG12D mutant mice grown in collagen are able to form branching structures whereas organoids grown in Matrigel do not.

The manuscript is well written and describes, in great detail, the observed branching process. Branching morphogenesis is interesting and the culture system established by the authors could prove valuable to study this process. However, both the practical and theoretical mathematical model need to be put in a biological context with additional controls to prove this point. Additionally, conclusive biological insights cannot be made due to the lack of essential controls.

To improve the biological relevance, I suggest the following: First controls in the form of healthy adult and embryonic murine pancreatic cells must be included. If the healthy cells are also able to undergo branching morphogenesis, a comparison of branching morphogenesis kinetics between healthy and tumour cells using the established mathematical model could be performed.

If the healthy cells do not undergo branching morphogenesis, it would support the authors’ claim that the branching process is PDAC specific. In that case, further biological insights can be gained by investigating whether the branching processes differs during different stages of tumorigenesis. For example, comparing organoids established from KrasG12D and KrasG12D; Tp53^{-/-} mutant mice.

Below I have listed a few points that I believe will help improve the manuscript.

Major points

- Controls in the form of healthy adult and embryonic pancreatic organoids are essential to determine whether the observed branching processes is indeed mimicking PDAC morphogenesis or whether it is a more common phenomenon of pancreatic cells in this culturing context.
- As the media condition differ from previously published studies of pancreatic organoids (more closely resembling that used for explants) the rationale for the choice of this media composition would be of interest. Can the authors explain how this medium was developed

- Further information about the culture of these organoids would be of interest. Can they be passaged following branching formation, and if yes how often?
- All the tables containing gene names need to have a higher resolution and larger or part of the supplementary material. They are almost unreadable and thus it is not possible to comment on the information therein.
- The authors want to show that proliferation, invasion, contraction and ion flux regulation are required for correct branching morphogenesis. However, the experimental design and chosen methods affect general organoid morphology as well. For example, organoids grown in Matrigel also swell when treated with forskolin, and thus this is not specific for branching.
- Fig 1F: histological sections from the primary murine tumour would be the best positive control, and transplantation of organoids grown in Matrigel are lacking. Especially since previously published studies have shown that murine and human PDAC organoids grown in Matrigel are able to recapitulate the primary tumours^{1,2}. If this is not the case it could be due to a difference in the media composition that the authors are using. This should then be tested.
- The authors make the following statement “Gene expression profiling of organoids shows a higher degree of epithelial differentiation in branched organoids cultured in collagen”. Based on what criteria do the authors evaluate epithelial differentiation and what data is supporting this claim?
- Fig 4C: (Inducible) shRNA/CRISPR experiments would strengthen the claim that MT1-MMP is important in PDAC organoid invasion.
- Could the authors elaborate on the idea behind the forskolin experiment. How is this specific to branching?
- The mathematical method for studying branching morphogenesis is interesting but lacks biological context to prove that it is useful.
- Fig 5D: early time points are missing and are necessary to determine how well the prediction fits with the real data. For this purpose, multiple different predictions are usually shown in the graph and the data plotted to see which model fits better.

Minor points

- Histology sections of primary PDAC samples would serve as a positive control and could strengthen the similarity between the morphology of the organoids, the transplanted organoid grafts and primary tumours
- Be consistent in the use of antibodies, sometimes K7 and other times K19 to stain for ductal cells
- Fig 1F: For the transplantation experiments, collection after 14 days is relatively short. Longer time points could be included. Did the authors try longer time points?
- Could the authors please clarify the link to the genetic context in their statement “Furthermore, our results indicate that tumor cells are able to execute inherent self-organizational programs induced by specific biophysical contexts, mirroring tumor morphogenesis to a remarkable degree, both at the genetic (Fig. 1D) and at the architectural levels (Fig. 1E, F)”.

- Fig 3C, inserts with a higher magnification would provide more details.
- Row 146: organoids were killed by Triton-X treatment. What is the rationale behind this?
- Row 169: I would avoid the term alveoli as it is most commonly associated with the lung. To avoid confusion microlumen might be better. Do the ALBOs (alveoli-like branched organoids) differ, biologically, to non-ALBOs? Could the authors comment on what they think the functional role of these microlumens is.
- Fig 4B: It is known that proliferation is required to fuel branching extension. Maybe the authors could use other inhibitors/ small molecules to try to find novel pathways involved in branching morphogenesis? The use of other proliferation inhibitors would strengthen the claim made by the authors.
- Row 306-308: please clarify the sentence. “Importantly, we found that this minimal model, fitted only on a live-short term time course between day 7 and 9 of organoid growth, could very well describe the entire time course from the Onset phase at day 1 to the thickening phase at day 10.”

References

1. Boj, S. F. et al. Organoid models of human and mouse ductal pancreatic cancer. *Cell* 160, 324–338 (2015).
2. Miyabayashi, K. et al. Intraductal Transplantation Models of Human Pancreatic Ductal Adenocarcinoma Reveal Progressive Transition of Molecular Subtypes. *Cancer Discov.* 10, 1566–1589 (2020).

Reviewer #2 (Remarks to the Author): Expert in pancreatic cancer development

In this study the authors established and characterized a tubular organotypic culture model of PDAC. As the authors stated, organoids recapitulating tubular structures of PDAC in vivo will be useful to study growth and differentiation of pancreatic cancers. The topic is interesting and highly relevant. Overall, the data are of good quality and the text is well written and clear. However, the current work relies on one mouse KC PDAC cell line, which limits potential biological relevance of the overall findings. Although PDAC cells in collagen gels mimic in vivo PDAC architecture, the authors did not provide any other evidence that the current collagen-based organoid model recapitulates physiologically relevant aspects of PDAC biology.

Major criticisms/ suggestions:

(1) What are the similarities and differences in formation of branched tubular structures between normal pancreatic ductal cells and tumor cells? Does the current collagen-based organoid system allow to study the differences in branching morphogenesis between normal and tumor cells?

(2) About 70% of all human PDAC cases harbor mutations in the TP53 tumor suppressor gene. Therefore, it would be important to assess tumor branching morphogenesis using KPC cell lines (Kras+/LSL-G12D; Trp53+/LSL-R172H; Pdx1-Cre).

(3) In Figure 4, assessing the effects of batimastat and Y-27632 on cell proliferation is required to exclude the possibility that disrupted morphogenesis by these inhibitors is simply due to decreased cell proliferation.

(4) Based on Figures 1F and 3C, there are noticeable cell death during lumen formation. Thus, it would make sense to incorporate cell death into the analytical model described in Figure 5.

Reviewer #3 (Remarks to the Author): Expert in branching morphogenesis and mathematical modelling

The authors report a protocol for culturing branched tubular pancreas-derived organoids. Branching of organoids was previously reported for organs like lungs (<https://dx.doi.org/10.3389/fcell.2021.631579>), kidneys (<https://doi.org/10.1016/j.celrep.2020.107963>) and other organs with branched epithelial, including mammary glands in an earlier publication from Bausch lab. The authors further combine imaging, RNA expression profiling and pharmacological interventions to dissect drivers of growth phases. The reported mechanisms contributing to branching morphogenesis e.g. cell proliferation, ECM remodeling and lumen swelling, in principle, are well established for other branched systems (<https://doi.org/10.3389/fcell.2021.671402>, <https://doi.org/10.1242/dev.184499>).

Next the authors report a minimalistic, but physically sound, mathematical model based on three basic process: cellular division, tip branching and tip migration speed. Importantly model also includes mass conservation law and empirically observed negative feedback between cellular division and branch width. The model could well describe cell number count from day 1 to day 10. To further validate the model the authors model phenotypic changes induced by pharmacological intervention, however no

output is provided neither in the main nor in supplementary texts and therefore model performance cannot be assessed.

The manuscript could be improved if the authors:

- put their contribution into the context of findings reported for other branched organoids;
- provide complete clear description of computational analysis;
- address technical comments listed below.

Additional comments:

Fig 4a depicts results of PCA. This analysis is based on $n=8$ samples with a number of features (parameters) $n \gg 10$. It is

Fig. 5a. Interestingly authors report a negative relationship between branch volumetric growth and branch width. Thus, potentially providing a rational for the observed negative relationship between generation number and branch width in some of the branched organs.

L259: "Building upon the results of RNA profiling and functional testing, we next aimed to probe quantitatively the basic principles governing the extension and thickening phase." It is not clear how model is related with RNA profiling data.

Inconsistent notation in the main and supp text e.g. w is used to denote branch width in the main text and r is used in the supp.

Model input parameters and computed outputs are described in detail in the supp, however the description in the main text is rather confusing and does not provide the reader with clear description of how model inputs were informed from the data.

l280: "Indeed, we found a strong and consistent negative relationship between branch volume growth, used as a proxy for local proliferation, and branch width (Fig. 5c)." Perhaps should be Fig. 5b.

Figure 5C: Experimental data is shown from day 1 till day 11. Whereas model prediction is not shown beyond day 10. Does growth in the model saturates as in the experiment?

Figure 5D: Experimental data is shown only in the narrow time range of day 6 to 10 and with from 19 to 22 μm , where as panels B and C show same variables in a wider range. Can depicted experimental data in Figure 5D be extended?

Lines 347-348: "A manipulation of these model parameters should yield changes relatable in experimental morphology, thus providing a starting point to rationalize the relation between gene regulation, growth parameters, and the resulting self-organization of the structure." No explicit or implicit connection between genes and biophysical model was described by the authors.

Figure S5C & D legends are hardly readable.

Response to the Reviewers "Spatiotemporal dynamics of self-organized branching in pancreas-derived organoids" in Nature Communications

We are grateful to the Reviewers for their constructive comments and suggestions, which we address in our Response below, and have implemented in the updated version of the manuscript. Thanks to their help, we believe to have considerably improved the quality of our submission.

Note to the Reviewers: unless explicitly stated otherwise, the references given for the figures in the present response, correspond to the references of the updated manuscript.

Reviewer #1 (Remarks to the Author): Expert in pancreatic cancer organoids and genomics

In the manuscript entitled "Spatiotemporal dynamics of self-organized branching in pancreas-derived organoids" the authors establish in vitro culture conditions that enable culturing of branching murine PDAC (pancreatic ductal adenocarcinoma) organoids. They use this system to show that organoids established from KrasG12D mutant mice grown in collagen are able to form branching structures whereas organoids grown in Matrigel do not.

The manuscript is well written and describes, in great detail, the observed branching process. Branching morphogenesis is interesting and the culture system established by the authors could prove valuable to study this process. However, both the practical and theoretical mathematical model need to be put in a biological context with additional controls to prove this point. Additionally, conclusive biological insights cannot be made due to the lack of essential controls.

To improve the biological relevance, I suggest the following: First controls in the form of healthy adult and embryonic murine pancreatic cells must be included. If the healthy cells are also able to undergo branching morphogenesis, a comparison of branching morphogenesis kinetics between healthy and tumour cells using the established mathematical model could be performed.

If the healthy cells do not undergo branching morphogenesis, it would support the authors' claim that the branching process is PDAC specific. In that case, further biological insights can be gained by investigating whether the branching processes differs during different stages of tumorigenesis. For example, comparing organoids established from KrasG12D and KrasG12D; Tp53^{-/-} mutant mice.

Below I have listed a few points that I believe will help improve the manuscript.

Major points

1) Controls in the form of healthy adult and embryonic pancreatic organoids are essential to determine whether the observed branching processes is indeed mimicking PDAC morphogenesis or whether it is a more common phenomenon of pancreatic cells in this culturing context.

While the focus of the manuscript is the morphogenesis of the PDAC, it certainly adds more insight to compare it to healthy organoid cultures. We cultured healthy adult pancreatic ductal cells (PDC) in our floating collagen assay, and observed that those cells only gave rise to cyst-like structures, under various media conditions (see Fig. S2c of the updated manuscript). To further probe whether the development of branched organoids was a feature of PDAC, we investigated the behaviour of tumour cells from a *Pdx1Cre; Kras^{G12D}; Trp53^{fl/fl}* mouse (KPC mouse). We now are able to report that KPC organoids develop into structures with strikingly similar morphology to organoids of a *Ptf1aCre; Kras^{G12D}* (KC mouse) origin, with a three-dimensional thick branched architecture bearing a lumen. Importantly, the parental primary tumor of the KPC mouse also displayed a well differentiated morphology with tubular structures, now shown in Fig. 1f. When we performed transcriptional profiling of KPC vs. KC organoids, KPC organoids displayed enriched signatures for proliferation pathways (Myc targets, E2F targets, etc.) and EMT, now shown in Fig. S2d-e. We added these results to the updated Fig. 1 and Fig. S2, and appended our analysis to the results section.

As for the embryonic pancreatic organoids, although this would be a very interesting application for our morphogenesis assay, due to time constraints we were not able to obtain governmental approval to perform timed matings to isolate embryonic pancreata.

2) As the media condition differ from previously published studies of pancreatic organoids (more closely resembling that used for explants) the rationale for the choice of this media composition would be of interest. Can the authors explain how this medium was developed

We thank the Reviewer for providing us with an opportunity to explain our rationale for medium selection. The classic organoid media composition, the Reviewer is referring to, favors epithelial differentiation and growth compared to our media. Branching morphogenesis, however, requires cells to be plastic and switch between different states, undergoing EMT and MET (Fig. 5a). Indeed, when we culture the KC cells in the more classical organoid media (1), we observed a pronounced reduction of branching and organoid complexity as well as increased formation of cystic organoids (see Figure below). We hypothesize that the cystic structures observed in these conditions are, in part, due to the presence of cholera toxin in the media, which acts similarly to Forskolin (see updated Fig. 5e and Fig. S3b).

In addition, our aim was to maintain similar culture condition as in previous work by Mueller *et al.* (1), where cell lines used in our manuscript have been characterized comprehensively in regards to

their biology and phenotypes in 2D.

Caption: Representative bright field images of collagen-grown KC organoids at day 13, cultured in PDC medium + 20 ng EGF.

3) Further information about the culture of these organoids would be of interest. Can they be passaged following branching formation, and if yes how often?

The Reviewer raises an important point. As indicated before, organoids can be propagated from 2D cultures of KC derived cells, from passages between 15-25 without any or major morphological variations. As per the Reviewer's suggestion, we performed series of passaging on branched organoids. Briefly, organoids were grown until day 13 in collagen, at which point the matrix was digested and the structures dissociated at the single cell level using 0.05 % trypsin/EDTA. Dissociated cells were then seeded again in a new collagen matrix, giving rise to secondary structures. Repeating this process on secondary structures at day 13, in turn, yielded tertiary structures.

To test again the potency of these cells, we performed these experiments in an extreme limiting dilution fashion, seeding 0.75 cells/gel and analysed them to quantify the ability of cells to form primary, secondary, and tertiary structures. We have added those results to Fig. S2g-i.

4) All the tables containing gene names need to have a higher resolution and larger or part of the supplementary material. They are almost unreadable and thus it is not possible to comment on the information therein.

We have improved the legibility of the tables.

5) The authors want to show that proliferation, invasion, contraction and ion flux regulation are required for correct branching morphogenesis. However, the experimental design and chosen methods affect general organoid morphology as well. For example, organoids grown in Matrigel also swell when treated with forskolin, and thus this is not specific for branching.

We thank the Reviewer for bringing to our attention a point that seems to have caused a misunderstanding.

We wish to state that we do not argue that the fundamental processes mentioned (proliferation, invasion, contraction, branching, ion flux regulation) are a feature unique and specific to PDAC

organoid development. Rather, since our study introduces a new type of organoid culture, our aim was to probe the effects of these fundamental processes, otherwise well-characterized in classic organoid culture conditions, in our branched organoids and, more specifically, in distinct phases of branched organoid development. We find that the spatio-temporal dynamics of those processes lead to the emergence of thick-branched, lumen-bearing, PDAC structures, which we describe in the paper.

As demonstrated in the current work, some developmental phases are more susceptible to manipulation, of e.g. ECM remodelling or contraction, than others. We believe these are all important findings as benchmark for future applications such as testing of drugs or genetic manipulations.

In detail, we would like to re-state and clarify that the scope of our perturbation experiments goes beyond “branching morphogenesis” *stricto sensu*, but that we rather investigate the entire organoid morphogenesis. Indeed, we argue that processes of proliferation, invasion, branching, contraction, and ion flux regulation, all play a role in the development of pancreas-derived branched three-dimensional structures connected by a single seamless lumen, ultimately resulting in the structures that we termed “ALBO” (Alveoli-Like Branched Organoids) in the initial version of the manuscript, now termed “TEBBO” (Terminal End Bud Branched Organoids). We find that those processes display different spatiotemporal dynamics in the course of development: for instance, there is intense branch extension and branching in the “Extension phase”, which stops in the “Thickening phase” (see updated Fig. 2a, Fig. S5a). Similarly, we for instance observe that the fraction of proliferative cells decreases over time (see updated Fig. 2c-e, Fig. 4d). To our knowledge, this is the first time the role of these processes in the morphogenesis of an organoid system has been reported in such detail.

The role of these processes is not limited strictly to the “branching” aspect of morphogenesis. In particular, the ion flux regulation appears to be mostly involved in the lumen formation of organoids (as shown in updated Fig. 5e, 5j), a hypothesis that we aimed to probe using forskolin, given that, as very correctly pointed out by the Reviewer, forskolin has been shown to induce swelling of cystic organoids (thus possessing a lumen) in Matrigel (2). We find rather, that it is the balance between those processes and their activity at the appropriate timepoints and with appropriate intensities, that seems to be crucial for the proper establishment of “TEBBO” structures. To support this claim, we used a series of inhibitors to interfere with those processes at various timepoints, as described in the subsections “Organoid proliferation drives branch extension”, “ECM remodeling is required for branching”, “Contraction is required for branched tubular structure formation”, “Ion flux drives lumen swelling”, and now shown in Fig. 5b-j.

Going back to the forskolin experiment, we find for instance that inducing excessive lumen swelling at early timepoints (by adding forskolin at Day 0) prevents the proper formation of “TEBBO” structures, instead giving rise to “cystic branched” and “cystic” structures. We added an additional reference for the forskolin treatment in the main text (2), to clarify the background behind our experiment.

6) Fig 1F: histological sections from the primary murine tumour would be the best positive control, and transplantation of organoids grown in Matrigel are lacking. Especially since previously published studies have shown that murine and human PDAC organoids grown in Matrigel are able to recapitulate the primary tumours^{1,2}. If this is not the case it could be due to a difference in the media composition that the authors are using. This should then be tested.

We thank the Reviewer for this comment. Indeed, the primary tissues are the best control. We amended Figure 1 to show a comparison between tissue sections of healthy pancreas, and the corresponding *Ptf1aCre; Kras^{G12D}* and *Pdx1Cre; Kras^{G12D}; Trp53^{fl/fl}* primary tumors, now shown in Fig. 1f.

In addition, we have previously shown that organoids, including patient-derived organoids, transplanted orthotopically into mice, recapitulate molecular and morphological features of the primary tumor (3). Our main goal in this manuscript was to recapitulate tubulogenesis, a key morphological feature of PDAC, *in vitro*. In this regard, the classic organoid conditions fail to mimic tumor architecture, as they solely yield cystic structures. In our transplantation assay, we therefore implanted single cells from 2D monolayers in order to observe morphogenesis in their natural environment, i.e. in the pancreas. Indeed, these cells were again able to display branching morphogenesis *in vivo* (now shown in Fig. 1Sb-c). Since our study emphasises an *in vitro* morphogenesis system, we did not perform a systematic comparison between Matrigel cultures organoids and collagen cultures organoids *in vivo* as these types of mouse experiments were not covered in our IACUC protocol.

7) The authors make the following statement “Gene expression profiling of organoids shows a higher degree of epithelial differentiation in branched organoids cultured in collagen”. Based on what criteria do the authors evaluate epithelial differentiation and what data is supporting this claim?

The bulk RNA-sequencing we performed reveals that organoids cultured in Matrigel possessed a transcriptome associated with a “basal-like” PDAC signature, whereas organoids grown in collagen possessed a transcriptome associated with a “classical” PDAC signature, as shown in the updated Fig. 1e. To further illustrate this claim, we have added to Fig. S2a the gene set enrichment analysis plot indicating the respective expression scores for the basal and classical signature, and to Fig. S2b a heatmap highlighting key genes involved in the epithelial to mesenchymal transition, differentially expressed between collagen- and Matrigel-grown organoids, as a supplementary hallmark from our GSEA plot indicating the epithelial differentiation in collagen grown branching organoids (Fig. 1e).

8) Fig 4C: (Inducible) shRNA/CRISPR experiments would strengthen the claim that MT1-MMP is important in PDAC organoid invasion.

We thank the Reviewer for their suggestion. In the manuscript we highlight the importance of an invasive process for branching morphogenesis. From our RNA seq experiments (Fig. 5c), we identified that matrix effectors such as cathepsin L (Ctsl), Timp1, Timp2, Furin and the membrane type 1 matrix metalloproteinase (Mmp-14) are significantly upregulated at early time points (day 7) of organoid formation. Invasion is strongly mediated by the MMP-driven collagen degradation, therefore, we focused on the effect from the inhibition experiments involving the broad-spectrum inhibitors marimastat and batimastat (now shown in Fig. 5c, Fig. S4), with the latter inhibitor additionally shown to target MMP-14 (4). This is consistent with literature reports on the role of MMP in collagen remodeling and branch extension (5), and in pancreatic cancer (6).

While we are certain that the Reviewer’s suggestion of shRNA/CRISPR experiments could indeed shed further light on the precise role of MT1-MMP in PDAC organoid invasion, and would make for an elegant follow-up experiment, we also believe that such an experiment is beyond the scope of what is claimed in the current manuscript. We consider that both the existing literature and the

experiments we performed provide sufficient backing for our claim that a broad spectrum of MMPs are involved in the invasion process.

9) Could the authors elaborate on the idea behind the forskolin experiment. How is this specific to branching?

We note that the Reviewers' concerns are also partly addressed in our reply to Major Point n°5, that complement the present reply. As pointed out by the Reviewer, forskolin has been shown to induce swelling of cystic organoids (thus possessing a lumen) in Matrigel (2).

In our study, the forskolin experiment was used to "demonstrate the importance of ion flux in the lumen formation process", as stated in the manuscript (line 248), and is therefore beyond the scope of a definition of "branching" *stricto sensu*, restricted to the budding of a branch, its extension, and the tip cell splitting giving rise to two daughter branches.

We find however that during the development of pancreatic organoids, the balance between the different processes (proliferation, invasion, branching, ion intake, etc.) and importantly their activity at the appropriate timepoints and with appropriate intensities, seems to be crucial for the proper establishment of "TEBBO" structures.

To test this assertion, the forskolin experiment was therefore designed to induce lumen formation at earlier timepoints than normal. Indeed, we observed that the addition of forskolin at day 0 or day 7 induces the formation of cystic branched and cystic phenotypes, as shown in Fig. 5e and quantified in Fig. 5j.

Regarding the cystic branched phenotype, which appears mainly when adding forskolin at Day 7, the branching process could proceed unhindered for seven days, but forskolin, by causing an excessive ion influx prior to the thickening of the branches, induces the formation of organoids displaying a large central lumen at the core with thin branches, instead of an "evenly spread" lumen across thick branches.

Adding forskolin at day 0 resulted in a cystic phenotype, where organoids develop into branchless cysts, thus deviating even further from the classic "TEBBO" phenotype appearing in untreated cases.

This experiment suggested that if (excessive) lumen swelling occurs prior to the normal extension or thickening phases, where the thick branched architecture is normally established, lumen formation can alter (in the cystic branched case) or even prevent (in the cystic case) branching morphogenesis.

It seems that the sentence "These results highlight the importance of a regulated swelling for the formation of thick-branched organoids connected by a seamless lumen." might let the reader think that the forskolin experiment might have been designed to specifically target branching, whereas our intent was to target regular lumen formation. We replaced the sentence by "These results highlight the importance of swelling regulated in time and intensity for the formation of a seamless lumen in thick branched organoids".

10) The mathematical method for studying branching morphogenesis is interesting but lacks biological context to prove that it is useful.

In order to clarify the importance of physical models in general and specifically of our model to the biology community, we expanded the Discussion part to better reflect how our study follows a growing body of literature aiming to understand the specificity of the branching morphogenesis

processes of each organ and organoid type (7, 8). We importantly note that minimal biophysical models using a limited set of local rules have been developed in an effort to find common framework to study various organs/organoids undergoing branching morphogenesis, in spite of the large underlying biological differences. To further highlight the usefulness of our analytical model, we performed additional morphometric characterization of untreated PDAC organoids, and compared this data to predictions of our model.

In particular, we believe that one important advantage of our experimental model system, compared to a number of *in vivo* models of branching, is the ability to get live measurements - in particular the relationship between branch width, tip invasion speed and cell growth between day 7 and day 9, which we show is sufficient input for the model to accurately predict the full spatio-temporal dynamics between day 1 and day 9. Additionally, in response to Reviewer #3, we extended our data and analysis significantly. We expanded the measurement of branch thickness, by measuring organoids at every day of development from day 2 to 9. To get more spatial information, we also performed these measurements both on “terminal”/elongating branches located at the edges of organoids and “non-terminal” branches (past a branching point, hence not elongating), finding differences that were well-recapitulated by our spatial model. Indeed, intuitively, the “older” branches should thicken earlier than the “younger” branches, but all should converge toward a branch thickness at which proliferation is abolished. To probe this prediction, we measured and compared the thickness of terminal branch segments possessing an active tip in fixed organoids (i.e. the “youngest” branches, of generation number N) with the thickness of non-terminal branch segments of generation number $N-1$, located between the two most recent branching points. Comparing the experimental data to the model data reveals excellent agreement, highlighting the ability of our model to perform intra-organoid, spatially-dependent prediction (see Fig. 4e).

Finally, we have now used the model to delve deeper on the question of not only the average properties of organoids, but also variance/heterogeneity from one organoid to the next. The organoid scientific community has recognized inter-organoid variability as one of the key challenges to investigate and address, notably for translational applications (9–11). We have shown that, while our system exhibits stereotypical development phases leading to the formation of three-dimensional branched structures bearing a single lumen, several organoid phenotypes can coexist in the culture dish (see Fig. S5A, S5B of the pre-revisions manuscript, and Fig. 4h of the updated manuscript). To analyze the variability present in our system, we identified the number of branches as a morphological hallmark of phenotype variability, and measured the branch number in organoids between day 2 and 10, revealing the extent of the diversity.

We are proud to report that, importantly, our model does not only capture the trend of branch increase over the developmental timecourse (see updated Fig. 4c, 4f), but also manages to capture the variance of the system (see updated Fig. 4g), even when considering strictly identical organoids at seeding time, only relying on the stochasticity of the development branching processes.

Moreover, the spatial simulations provide an additional confirmation of the power of our model, by qualitatively predicting the existence of “short thick-branched” phenotypes and the “long thinner-branched” phenotypes (see Fig. S5A, S5B of the pre-revisions manuscript, and Fig. 4h in the revised manuscript).

We have updated in consequence the main text (section “A minimal theoretical model captures branching organoid morphogenesis”) as well as the supplementary text. We switched the order of Fig. 4 and Fig. 5 of the pre-revisions manuscript, as well as the order of the “A minimal theoretical model captures branching organoid morphogenesis” and “Distinct dynamically regulated

transcriptional programs orchestrate branching organoid development” paragraphs to better reflect our rationale.

11) Fig 5D: early time points are missing and are necessary to determine how well the prediction fits with the real data. For this purpose, multiple different predictions are usually shown in the graph and the data plotted to see which model fits better.

We note that we also address in detail the Reviewer’s concern in point 10), that this answer complements.

We agree that extending the range of data shown to early timepoints is beneficial to the figure and we are grateful for the Reviewer’s comment, as it motivated us to go beyond the originally presented data and analysis, thereby allowing us to considerably improve our claims.

We have measured additional organoids every day from day 2 to 9 and have amended the plot in consequence, now shown in Fig. 4e. We also plotted our model prediction on the same graph, and note the good agreement between the experimental data, and the model prediction.

We have updated in consequence the main text (section “A minimal theoretical model captures branching organoid morphogenesis”) as well as the supplementary text.

Minor points

12) Histology sections of primary PDAC samples would serve as a positive control and could strengthen the similarity between the morphology of the organoids, the transplanted organoid grafts and primary tumours

As suggested by the Reviewer, we performed haematoxylin and eosin staining of primary tissue sections of wild type pancreas, of tumours in *Ptf1aCre; Kras^{G12D}* mice, and of tumours in *Pdx1Cre; Kras^{G12D}; Trp53^{fl/fl}* mice. The results are now shown in Fig. 1f.

13) Be consistent in the use of antibodies, sometimes K7 and other times K19 to stain for ductal cells

Unfortunately, the Cytokeratin 19 antibody (Troma-III DSHB) does not work in whole-mount organoid stainings. For consistency, we have removed the K19 staining from Figure 1 and transferred it to the updated Fig. S1b.

14) Fig 1F: For the transplantation experiments, collection after 14 days is relatively short. Longer time points could be included. Did the authors try longer time points?

The transplantation experiment was designed to check whether transplanted PDAC cells retained *in vivo* the ability to invade and form structure they displayed *in vitro*, in the collagen assay. For this reason, we purposefully kept the collection timeframe similar to the *in vitro* experiment, in which organoids were studied for thirteen days. Based on different scientific questions, we have previously performed later time points collections, for example to study tumour progression and metastasis. We frequently observed a certain degree of increased tumour necrosis. This is an important confounding factor for tumour morphogenesis we wanted to exclude in this study.

15) Could the authors please clarify the link to the genetic context in their statement “Furthermore, our results indicate that tumor cells are able to execute inherent self-organizational programs induced by specific biophysical contexts, mirroring tumor morphogenesis to a remarkable degree, both at the genetic (Fig. 1D) and at the architectural levels (Fig. 1E, F)”.

We have amended the sentence and replaced “genetic” by “transcriptional”. We thank the Reviewer for pointing out the typo.

16) Fig 3C, inserts with a higher magnification would provide more details.

We believe that the exporting process may have degraded the image quality in the pre-submission manuscript. We have updated the images with a much higher quality to provide more details. The aSMA/Caspase 3/DAPI panel has been updated with a higher magnification.

17) Row 146: organoids were killed by Triton-X treatment. What is the rationale behind this?

As illustrated in Fig. 2b, we observed that pancreatic organoids in collagen interacted mechanically with the surrounding extra-cellular matrix (ECM), notably during the Onset and the Thickening phases. Moreover, we identified that MMP-driven collagen degradation allowed branches to extend during the Extension phase (Fig. 5c, Fig. 5h). Observing the structure of the collagen matrix *via* reflection microscopy revealed that organoids were indeed remodelling their environment: thickening branches can for instance align collagen fibres in front of them and appear to be surrounded by a layer of collagen, reminiscent of the findings reported in mammary gland organoids (5) (Fig. 3a). A question then arose: is this remodelling elastic – in which case fibres would relax as soon as forces stop being exerted –, or is it plastic – indicating a permanent mechanical deformation of the matrix –? To check this, we used Triton-X to degrade the cell membranes, effectively killing the organoids, dissociating them and abolishing the forces exerted on the matrix, leaving the collagen untouched. As indicated in Fig. 3b and Fig. S1d-e, the architecture of the collagen surrounding the organoids is preserved, hinting at a permanent, plastic deformation of the ECM.

Following the Reviewer’s comment, we have expanded on the rationale of Triton-X use in the main text, section “Thickening phase”.

18) Row 169: I would avoid the term alveoli as it is most commonly associated with the lung. To avoid confusion microlumen might be better. Do the ALBOs (alveoli-like branched organoids) differ, biologically, to non-ALBOs? Could the authors comment on what they think the functional role of these microlumens is.

We share the Reviewer’s concern about using the term “alveoli”. As we already used the term microlumen, inspired by Dumortier et al. (12), to describe the small lumen structures that nucleate along a branch, as shown in Fig. 3e for instance, we propose replacing the term “alveoli” by “terminal end bud” which we believe describes more accurately the structure. In the current version of the manuscript, we replaced “Alveoli-like Branching Organoids (ALBO)” by “Terminal End Bud

Branched Organoids (TEBBO)".

Whether ALBOs differ from non-ALBOs structures is a very interesting question. The majority of organoids formed in our assay resemble structures with terminal end buds. Therefore, our analyses were focused on these organoids. Whether, organoids lacking terminal endbuds (non-TEBBO) display a different biology, is a question we cannot answer at this point. They however are derived from the same parental PDAC line. Given the exponential and stochastic nature of branching according to our mathematical model, this could suggest that the stochasticity might be a "sufficient" factor to explain phenotypical variance.

As for the functional role of the terminal end buds, we believe they present the most mature phenotype in our system. During branching morphogenesis (between day 5 and 7), tip cells are highly invasive as the structure expands. Once this process has completed, invadopodia disappear and terminal end buds are formed.

19) Fig 4B: It is known that proliferation is required to fuel branching extension. Maybe the authors could use other inhibitors/ small molecules to try to find novel pathways involved in branching morphogenesis? The use of other proliferation inhibitors would strengthen the claim made by the authors.

While we share the Reviewer's interest in investigating novel pathways involved in branching morphogenesis, we believe that this endeavour would warrant its own study, and falls outside the scope of our manuscript.

20) Row 306-308: please clarify the sentence. "Importantly, we found that this minimal model, fitted only on a live-short term time course between day 7 and 9 of organoid growth, could very well describe the entire time course from the Onset phase at day 1 to the thickening phase at day 10."

As detailed in the Supplementary Text, section II. A, our model made the assumption that a negative feedback, further assumed to be linear, exists from the thickness of the branch on proliferation. We assumed a feedback of the form $K(t) = k_d(1 - w(t)/w_o)$, where $K(t)$ is the rate of volumetric growth, used as a proxy to estimate the proliferation in the organoid at time t , k_d is the maximal proliferation rate, $w(t)$ is the branch width at time t , and w_o is the maximal width, at which proliferation is inhibited (by contact inhibition for instance).

In order to check this assumption, we acquired live data between day 7 and 9 of organoid growth, quantifying the tip speed, branch length and branch width for a number of randomly selected terminal branches in the organoid. Measuring this data allowed us to estimate values for k_d and w_o , that we use as fitted parameters in our model. Moreover, we could determine from the live imaging data that there was no apparent correlation between the invasion speed of a branch and its width, which was another important assumption of our model.

From these checks and these fits, performed using data from a voluntarily limited timeframe, our model was able to make quantitative statements, for instance spontaneously reproducing the evolution of the cell number over a much longer time window (between day 1 and day 9, see Fig. 4d), and notably the switch from an exponential proliferation in the Onset phase, to a plateauing growth in the later stages, consistently with the data shown in Fig. 2c-e.

According to the Reviewer's comment, we expanded on our reasoning in the main text section "A

minimal theoretical model captures branching organoid morphogenesis “ and in the Supplementary text, section II. Furthermore, we added a summary of of parameter estimation and simulations input (Supplementary text, section III. A) to clarify which parameters were selected, and how.

Reviewers' references

1. Boj, S. F. et al. Organoid models of human and mouse ductal pancreatic cancer. *Cell* 160, 324–338 (2015).
2. Miyabayashi, K. et al. Intraductal Transplantation Models of Human Pancreatic Ductal Adenocarcinoma Reveal Progressive Transition of Molecular Subtypes. *Cancer Discov.* 10, 1566–1589 (2020).

Reviewer #2 (Remarks to the Author): Expert in pancreatic cancer development

In this study the authors established and characterized a tubular organotypic culture model of PDAC. As the authors stated, organoids recapitulating tubular structures of PDAC *in vivo* will be useful to study growth and differentiation of pancreatic cancers. The topic is interesting and highly relevant. Overall, the data are of good quality and the text is well written and clear. However, the current work relies on one mouse KC PDAC cell line, which limits potential biological relevance of the overall findings. Although PDAC cells in collagen gels mimic *in vivo* PDAC architecture, the authors did not provide any other evidence that the current collagen-based organoid model recapitulates physiologically relevant aspects of PDAC biology.

We firmly believe that a quantitative understanding of the morphogenetic process of a PDAC model system, including the differentiation processes is of outmost relevance as a benchmark for any further studies. To accommodate the Reviewer's concerns from the viewpoint of biological relevance, we have performed experiments studying branching morphogenesis in organoids derived from primary cells of 3 different KC mice (see Fig. S2f) and also a KPC mouse (Fig. 1f). In addition to mimicking the *in vivo* PDAC architecture, we provide comprehensive molecular and functional data relevant to PDAC biology. Specifically, we provide a model system that allows to study cellular plasticity during morphogenesis. Transcriptional profiling revealed key pathways driving PDAC biology such as invasion indicated by EMT and ECM remodelling signatures.

In addition to these general remarks, we provide a detailed point-by-point response below.

Major criticisms/ suggestions:

(1) What are the similarities and differences in formation of branched tubular structures between normal pancreatic ductal cells and tumor cells? Does the current collagen-based organoid system allow to study the differences in branching morphogenesis between normal and tumor cells?

As suggested by the Reviewer, we put healthy pancreas ductal cells to the test in our collagen assay. This experiment revealed that, in stark contrast to tumour cells which form highly-branched three-dimensional structures, healthy cells are only able to form cysts. We have amended Figure 1 to reflect these results, now shown in Fig. 1f.

(2) About 70% of all human PDAC cases harbor mutations in the TP53 tumor suppressor gene. Therefore, it would be important to assess tumor branching morphogenesis using KPC cell lines (Kras+/LSL-G12D; Trp53+/LSL-R172H; Pdx1-Cre).

As per the Reviewer's suggestion, we investigated the behaviour of cells isolated from a *Pdx1Cre*; *Kras*^{G12D}; *Trp53*^{fl/fl} mouse tumour in the floating collagen assay. We observed that these cells were capable of forming three-dimensional thick branched structures bearing a lumen, highly reminiscent of the organoids originating from the *Ptf1aCre*; *Kras*^{G12D} cells. We have amended Figure 1 to reflect those results, now shown in Fig. 1f.

In addition, we compared their gene expression by RNA seq analysis and provide new GSEA illustrating hallmarks that are differentially regulated in KPC vs KC derived organoids, now shown in Fig. S2d, S2e.

(3) In Figure 4, assessing the effects of batimastat and Y-27632 on cell proliferation is required to exclude the possibility that disrupted morphogenesis by these inhibitors is simply due to decreased cell proliferation.

We thank the Reviewer for their comment. We investigated the effect of batimastat and Y-27632 on the fraction of proliferation-capable cells by treating organoids at seeding time, and evaluating the ratio between Ki67-positive cells over DAPI-positive cells (the “proliferation index”), at a high proliferative state (day 7) and at a low proliferative state (day 13) of development, comparing treated and non-treated organoids. For treated-organoids, treatment was renewed for every medium change (as in the Materials and Methods section, D0, D3, D5, D7, D9, D11), with batimastat added at a concentration of 10 μ M, and Y-27632 at a concentration of 5 μ M. The results of this experiment is now depicted in Fig. S5g-h.

During the high proliferative state (day 7) we observe that the proliferation index is not significantly changed by treatments, while at the low proliferative state (day 13) the addition of batimastat significantly increases the ratio of Ki67 positive cells, suggesting that organoids subjected to a long-term batimastat treatment might maintain a higher proliferation capacity. On the contrary, treatment with Y-27632 has no significant effect. This suggests that the physical effects that we report in Fig. 5 do not directly arise from a decrease in the proliferation capability of cells, but rather from distinct mechanisms.

Indeed, we have shown that treating the organoids with batimastat prevents the elongation of branches, notably during the Extension phase (Fig. 5c, Movie S12), which, in correlation with the RNA-seq results (Fig. 5c), seem to highlight the importance of MMPs for the degradation of collagen, and thus the invasion of branches.

Similarly, we have shown that the addition of the Y-27632 inhibitor to the system appears to lead to a loss of contractility of the cells, as shown in Fig. 5d and Fig. 5i, and as previously reported in the literature (5), preventing in the end the establishment of thick-branched structures bearing a single seamless lumen.

(4) Based on Figures 1F and 3C, there are noticeable cell death during lumen formation. Thus, it would make sense to incorporate cell death into the analytical model described in Figure 5.

We note with interest the Reviewer’s suggestion, and believe that, for a model aiming to recapitulate the lumen formation dynamics, the processes of extrusion and apoptosis, contributing to the hollowing of the branches, would be extremely helpful to incorporate.

We would like nonetheless to re-state that the scope of our presented analytical model is narrower, as we deliberately focused on describing the phases of development ranging from the Onset to the (early) Thickening – i.e. from day 1 to day 9. Those early phases, where “branching morphogenesis” *stricto sensu* (i.e., the formation of the branched architecture) is occurring, appear to be minimally describable using the parameters of cell growth, branch extension, and of tip-splitting for branching. However, we note that strictly speaking, our relationship between branch width and growth is more general than proliferation, as it’s purely based on “volumetry”: although most of branch volume changes are due to proliferation, cell death would also enter into it. We clarified this in the Supplementary Theory Note (section II. A).

Reviewer #3 (Remarks to the Author): Expert in branching morphogenesis and mathematical modelling

The authors report a protocol for culturing branched tubular pancreas-derived organoids. Branching of organoids was previously reported for organs like lungs (<https://dx.doi.org/10.3389/fcell.2021.631579>), kidneys (<https://doi.org/10.1016/j.celrep.2020.107963>) and other organs with branched epithelial, including mammary glands in an earlier publication from Bausch lab. The authors further combine imaging, RNA expression profiling and pharmacological interventions to dissect drivers of growth phases. The reported mechanisms contributing to branching morphogenesis e.g. cell proliferation, ECM remodeling and lumen swelling, in principle, are well established for other branched systems (<https://doi.org/10.3389/fcell.2021.671402>, <https://doi.org/10.1242/dev.184499>).

Next the authors report a minimalistic, but physically sound, mathematical model based on three basic process: cellular division, tip branching and tip migration speed. Importantly model also includes mass conservation law and empirically observed negative feedback between cellular division and branch width. The model could well describe cell number count from day 1 to day 10.

0A) To further validate the model the authors model phenotypic changes induced by pharmacological intervention, however no output is provided neither in the main nor in supplementary texts and therefore model performance cannot be assessed.

We thank the Reviewer for their input, and for prompting us to improve and clarify the predictive power of the model outputs. Before turning to pharmacological conditions, we have performed extensive new experimental quantifications and theoretical analysis to better validate the model (some of which also in response to subsequent questions of the referee). In particular, we have:

- Complemented the previous measurements of cell number count in time with a systematic measurement across timepoints of branch number count (updated Fig. 4f). As predicted from the model with a single branching rate k_b during the branching phase (day 2 to day 9), we found that the average branch number in time was well-fitted by an exponential function.
- Investigated not only the average properties of branched organoids, but also their variance. Indeed, another key prediction of our model of stochastic branching is that the variance in total branch number should also grow exponentially in time, and be of comparable magnitude to the average in total branch number. Importantly, comparing this prediction with the data revealed good quantitative agreement, arguing that most of the heterogeneity in branching organoid size (in terms of branch number) could arise from the intrinsic stochasticity of branching at a given Poisson rate. This has been added as a new panel in Fig. 4g.
- Improved the temporal resolution of our measurements of branch width to every day between day 2 and day 9. Furthermore, to give further insights into the spatial regulation of branched organoids morphology (also in response to point 2 of the review below), we have separately measured and plotted the average width of “terminal” branches (with a growing tip) and of branches past a given branch point. Indeed, our spatial model predicts that terminal branches should converge towards a smaller width than non-terminal branches,

given that they are constantly “thinned” by elongation, while non-terminal branches can freely reach the width at which their growth is inhibited. Importantly, this was in full agreement with our new data – which has been added as a panel in Fig. 4e. We note that we stopped the quantifications in all of these datasets at day 9, given the gradual process of contraction and lumen formation that occurs around these time points (already present to a small degree around day 8 and day 9, but becoming dominant after day 10) – something out of scope of the current modelling.

- We switched the order of Fig. 4 and Fig. 5 of the pre-revisions manuscript, as well as the order of the “A minimal theoretical model captures branching organoid morphogenesis” and “Distinct dynamically regulated transcriptional programs orchestrate branching organoid development” paragraphs to better reflect our rationale.

The manuscript could be improved if the authors:

OB) put their contribution into the context of findings reported for other branched organoids

We thank the Reviewer for bringing to our attention the various references mentioned in their reports. We have amended our reference list and our Discussion paragraph in consequence, to better reflect the research landscape in the field of branched organoid.

OC) provide complete clear description of computational analysis;

As per the Reviewer’s suggestion, we added information to the technical Supplementary Note, in particular in relation to the new analysis and quantifications described above. Given our new emphasis of spatial aspects of branching morphology, we have also expanded our description of the spatial model (which gives similar predictions as the analytical model, but with additional layers of descriptions, notably for the variance in branch number and the difference between terminal and non-terminal branches).

- address technical comments listed below.

Additional comments:

1) Fig 4a depicts results of PCA. This analysis is based on $n=8$ samples with a number of features (parameters) $n \gg 10$. It is

We do not fully understand the point raised by the Reviewer here. Principal component analysis for groups of samples was computed using the *prcomp* function from the R *stats* package with default parameters. The input data matrix contained variance stabilized gene expression with the respective samples in rows and the 2000 most variable genes among these samples in columns. The median absolute deviation (MAD) was used to determine the variability of a given gene. We refer the

Reviewer to the Material and Methods sections “RNA-sequencing”, “Differential gene expression analysis”, “Gene Set Enrichment Analysis”.

2) Fig. 5a. Interestingly authors report a negative relationship between branch volumetric growth and branch width. Thus, potentially providing a rational for the observed negative relationship between generation number and branch width in some of the branched organs.

We thank the Reviewer for their comment and have expended the Discussion section to relate our findings to observations in other model systems. As described above, we have also expanded our analysis and theoretical predictions on this relationship between generation number and branch width, and comment further on this in the discussion, and in the Supplementary Text (in particular in section III. D. Simulation output: spatial dynamics of tip and branch morphometrics”).

3) L259: “Building upon the results of RNA profiling and functional testing, we next aimed to probe quantitatively the basic principles governing the extension and thickening phase.” It is not clear how model is related with RNA profiling data.

We agree with the Reviewer that our initial wording was unclear. The RNA sequencing, itself informing the functional experiments performed using inhibitors, evidenced the role of three processes fundamental for the establishment of a branched organoid: proliferation (as evidenced in Fig. 5B, Fig. S2A in the pre-revision manuscript, Fig. 5b and Fig. S3a in the updated manuscript), branching – also partly mediated by proliferation (as evidenced in Fig. 5B, Fig. S2A, Fig. S4D in the pre-revision manuscript, Fig. 5b, Fig. S3a, Fig. S5d in the updated manuscript) –, and invasion – mediated by MMP (as evidenced in Fig. 5C, Fig. S3 in the pre-revision manuscript, Fig. 5c, Fig. S4 in the updated manuscript) –. Given those experiment observations, we reasoned that those three processes could form the base for a first minimal biophysical model.

We reworded the sentence to clarify it.

4) Inconsistent notation in the main and supp text e.g. w is used to denote branch width in the main text and r is used in the supp.

We thank the Reviewer for catching the typo, and have harmonized the notations to emphasize branch width.

5) Model input parameters and computed outputs are described in detail in the supp, however the description in the main text is rather confusing and does not provide the reader with clear description of how model inputs were informed from the data.

We have reworked our model description for clarification, and have added in particular a summary for the input parameters of the simulations (see section III.A. Summary of parameter estimation and simulation inputs of the SI Theory Note).

6) I280: “Indeed, we found a strong and consistent negative relationship between branch volume growth, used as a proxy for local proliferation, and branch width (Fig. 5c).” Perhaps should be Fig. 5b.

We thank the Reviewer for catching the typo, and have amended the reference in consequence. The graph can now be found in Fig. 4b.

7) Figure 5C: Experimental data is shown from day 1 till day 11. Whereas model prediction is not shown beyond day 10. Does growth in the model saturates as in the experiment?

We thank the Reviewer for providing us with the opportunity to clarify the scope of our model. In the model, there is a cross over between two regimes of exponential growth: a fast one where the slope is the rate of cell division k_d^0 (when average branch width is small and thus the feedback on growth is weak), and a slower – although still exponential phase – where the slope of growth is driven by the branching rate k_b (as branching becomes the limiting process of growth while cell division/branch width is close to full inhibition of growth).

Although the data does follow the gradual slowing down of growth predicted in the model, this slowdown is even more pronounced in the data starting around day 9 as tips start rounding up, stop their elongation/branching and the phase of lumen formation begins. Thus, we intentionally restricted our data range to the modelling of the branching and thickening dynamics, occurring in general between day 1 and 9. Lumen formation involves markedly different processes, such as fluid intake, fluid pressure, or cell shape changes, which are outside of the scope of the current model. In particular, the present manuscript uses the width of the branches as a proxy to estimate the maximal division rate k_d : this approximation is valid when the branches are not hollow; it however loses its validity when the lumen start swelling, and when fluid intake starts contributing to the thickness of the branch, independently of whether proliferation is actually taking place.

To clarify the scope of our model, we have amended the main text section “A minimal theoretical model captures branching organoid morphogenesis”, and the Supplementary Text section II. “Parameter estimation and model predictions”.

8) Figure 5D: Experimental data is shown only in the narrow time range of day 6 to 10 and with from 19 to 22 μm , where as panels B and C show same variables in a wider range. Can depicted experimental data in Figure 5D be extended?

As described above, we have performed additional quantifications of these quantities, as well as a number of other metrics, at every day of development between day 2 and 9.

We restricted our analysis to day 9 instead of day 10 as previously shown, as we judged that the strong contraction and the beginning of lumen formation observed at day 10 in the Thickening phase (as seen in Fig. 2a-b) corresponds to a physical event falling outside of the scope of the model. We have updated Fig. 4 with a subfigure (Fig. 4e) showing side by side the experimental data and the model prediction, where we note the excellent agreement between the two.

9) Lines 347-348: “A manipulation of these model parameters should yield changes relatable in experimental morphology, thus providing a starting point to rationalize the relation between gene

regulation, growth parameters, and the resulting self-organization of the structure.” No explicit or implicit connection between genes and biophysical model was described by the authors.

We have edited the sentence to make it clear that it was meant as an “outlook” statement, as we believe that follow-up studies on organoids, using the depicted combination of sequencing data, live-cell imaging, and biophysical modelling, will be able to relate together the genes, growth parameters, and the dynamics of cell organization.

10) Figure S5C & D legends are hardly readable.

We have reworked the legends to improve legibility, now shown in Fig. S6.

Bibliography

1. S. Mueller, T. Engleitner, R. Maresch, M. Zukowska, S. Lange, T. Kaltenbacher, B. Konukiewitz, R. Öllinger, M. Zwiebel, A. Strong, H. Y. Yen, R. Banerjee, S. Louzada, B. Fu, B. Seidler, J. Götzfried, K. Schuck, Z. Hassan, A. Arbeiter, N. Schönhuber, S. Klein, C. Veltkamp, M. Friedrich, L. Rad, M. Barenboim, C. Ziegenhain, J. Hess, O. M. Dovey, S. Eser, S. Parekh, F. Constantino-Casas, J. De La Rosa, M. I. Sierra, M. Fraga, J. Mayerle, G. Klöppel, J. Cadiñanos, P. Liu, G. Vassiliou, W. Weichert, K. Steiger, W. Enard, R. M. Schmid, F. Yang, K. Unger, G. Schneider, I. Varela, A. Bradley, D. Saur, R. Rad, Evolutionary routes and KRAS dosage define pancreatic cancer phenotypes. *Nature*. **554**, 62–68 (2018).
2. S. F. Boj, A. M. Vonk, M. Statia, J. Su, J. F. Dekkers, R. R. G. Vries, J. M. Beekman, H. Clevers, Forskolin-induced Swelling in Intestinal Organoids: An In Vitro Assay for Assessing Drug Response in Cystic Fibrosis Patients. *JoVE*, e55159 (2017).
3. Z. Dantes, H.-Y. Yen, N. Pfarr, C. Winter, K. Steiger, A. Muckenhuber, A. Hennig, S. Lange, T. Engleitner, R. Öllinger, R. Maresch, F. Orben, I. Heid, G. Kaissis, K. Shi, G. Topping, F. Stögbauer, M. Wirth, K. Peschke, A. Papargyriou, M. Rezaee-Oghazi, K. Feldmann, A. P. G. Schäfer, R. Ranjan, C. Lubeseder-Martellato, D. E. Stange, T. Welsch, M. Martignoni, G. O. Ceyhan, H. Friess, A. Herner, L. Liotta, M. Treiber, G. von Figura, M. Abdelhafez, P. Klare, C. Schlag, H. Algül, J. Siveke, R. Braren, G. Weirich, W. Weichert, D. Saur, R. Rad, R. M. Schmid, G. Schneider, M. Reichert, Implementing cell-free DNA of pancreatic cancer patient-derived organoids for personalized oncology. *JCI Insight*. **5** (2020), doi:10.1172/jci.insight.137809.
4. A. Winer, S. Adams, P. Mignatti, Matrix Metalloproteinase Inhibitors in Cancer Therapy: Turning Past Failures Into Future Successes. *Mol. Cancer Ther.* **17**, 1147–1155 (2018).
5. B. Buchmann, L. K. Engelbrecht, P. Fernández, F. P. Hutterer, M. K. Raich, C. H. Scheel, A. R. Bausch, Mechanical plasticity of collagen directs branch elongation in human mammary gland organoids. *Nat. Commun.*, 1–10 (2021).
6. V. Ellenrieder, B. Alber, U. Lacher, S. F. Hendler, A. Menke, W. Boeck, M. Wagner, M. Wilda, H. Friess, M. Büchler, G. Adler, T. M. Gress, Role of MT-MMPs and MMP-2 in pancreatic cancer progression. *Int. J. Cancer*. **85**, 14–20 (2000).
7. C. Lang, L. Conrad, D. Iber, Organ-Specific Branching Morphogenesis. *Front. Cell Dev. Biol.* **9**, 1–17 (2021).
8. K. Goodwin, C. M. Nelson, Branching morphogenesis. *Development*. **147** (2020), doi:10.1242/dev.184499.
9. B. L. LeSavage, R. A. Suhar, N. Broguiere, M. P. Lutolf, S. C. Heilshorn, Next-generation cancer organoids. *Nat. Mater.* **21**, 143–159 (2022).
10. M. Huch, J. A. Knoblich, M. P. Lutolf, A. Martinez-Arias, The hope and the hype of organoid research. *Development*. **144**, 938–941 (2017).
11. Advances in Organoid Technology: Hans Clevers, Madeline Lancaster, and Takanori Takebe. *Cell Stem Cell*. **20** (2017), doi:https://doi.org/10.1016/j.stem.2017.05.014.
12. J. G. Dumortier, M. Le Verge-Serandour, A. F. Tortorelli, A. Mielke, L. de Plater, H. Turlier, J.-L. Maître, Hydraulic fracturing and active coarsening position the lumen of the mouse blastocyst. *Science (80-)*. **365**, 465–468 (2019).

REVIEWER COMMENTS

Reviewer #1 (Remarks to the Author):

The authors have significantly improved the manuscript and addressed all but one of my concerns.

My main concern is regarding the WT controls. The materials and methods section gives me the impression that WT cells were cultured differently (media and coated plate vs flask) compared to the mutant cells prior to being subjected to the floating collagen assay. As differences in culture conditions have an effect of cell state and can thereby affect the ability of the cells to form branching structures.

Could the authors please clarify this. If different culture conditions are used, I believe the WT control might not fully serve its purpose. Ideally the experiment should be repeated using the same conditions for the control and mutant cells. Though, further clarification and making this obvious to the reader would also be satisfactory.

This could be achieved by:

- A schematic overview could be added in Fig1 that shows the procedures for the mutant and the wildtype cells (this would make it clear to the readers -as many do not carefully read the materials and methods section)
- Materials and methods section could be changed to have one section for the healthy cells and one for the mutant cells – to make it clear to the reader that there is indeed a significant difference.

In fig1 or as a supplementary I would like to see a zoomed-out image (of a well, or significant portion of a well) containing WT organoids as well a well containing branching organoids.

This is to show that the observed phenotypic difference holds true on a population level and not only for a single organoid. to give the reader an idea what the population looks like.

If these things are addressed, I am satisfied with the revision. The authors have greatly improved the manuscript and I appreciate the extended explanation, data and analysis with regards to their model as it highlights its robustness and applicability.

Reviewer #3 (Remarks to the Author):

In the revised manuscript the authors have extended quantitative experimental data and computational modelling. Notably experiment and model predictions are largely in agreement in the absence of pharmacological perturbations, whereas modelling of branching in the presence of pharmacological inhibitors require further clarification and/or refinement. The authors have also improved the manuscript flow.

Specific comments:

Section, A minimal theoretical model captures branching organoid morphogenesis, Fig 2E, Fig 4b I. 212 – 222.

>> The author observed a negative relationship between cell proliferation rate and time as well as branch volume growth and branch width. The authors attribute these observations to negative mechanical feedback, whereas models has been also proposed to describe growth inhibition based on diffusional limitations e.g. <https://doi.org/10.1093/jb/mvu087>, <https://doi.org/10.1098/rsob.130088>. The authors should either provide convincing arguments why they ruled out diffusional limitations or refer readers to both as plausible mechanisms.

L 251. “see Supplementary Note for details, and Fig. 4d-h, Fig. S5 for typical simulation outputs”

>> Fig. S5 does not contain any simulation output

L 255 – 257. “Secondly, our model also predicted that the variability and variance in branch number across organoids should also grow exponentially, and be of the same order of magnitude as the average itself – characteristic of a stochastic branching process. Strikingly, this described very well the data (Fig. 4g), arguing that most of the heterogeneity in branching organoid size could possibly arise from the intrinsic stochasticity of branching, rather than intrinsically different average rates of branching.”

>> proportional error is characteristic for many processes. If authors would like to rule out a particular mechanism based, it’s incompatibility with the variance to the mean must be explicitly demonstrated.

Fig. 4e.

>>Experimental data suggests that the average branch thickness continues to increase over 9 day period, whereas the model predicts flattening of the average thickness after 4 days. A discussion of this apparent discrepancy will benefit the reader.

L 269: “Finally, the model can be tested by observing the influence of batimastat treatment (a matrix metalloproteinase inhibitor, further described in the following section), which was found to abolish tip invasion speed ($v_0 \approx 0$). While branch volumetric growth is also perturbed in this condition (-77%), it remains active to a degree, so that by simple conservation law, we predict it to cause marked thickening of organoids (Fig. S6b-c), which was observed experimentally (Fig. 5c, Fig. S6f), as described above.”

>> Model sensitivity analysis indicates that the invasion speed has virtually no effect branch thickness (Fig Sup 6 b). This is contrary to the detailed experimental observation in presence of MMP inhibitors (Fig. S4)

L. 274: “However, if proliferation is inhibited, as with an aphidicolin treatment (a proliferation inhibitor, further described in the following section), the model predicts thinning of the organoid in time.”

>> Are these modelling results explicitly depicted in the publication?

Reviewer #4 (Remarks to the Author): Expert in pancreatic cancer and tumour microenvironment

The current manuscript by Dr Bausch and colleagues described a new method to self organizing, branching organoids that replicate the structure of pancreatic cancer; the authors define different phases of the organoid organization process, each defined by its own gene expression pattern. Further, the authors developed a model to describe the process of branching morphogenesis in the organoids. Overall, the authors developed a new #D model to mimic growth and 3D organization of pancreatic cancer, complementing existing models in the field.

In this revised application, the authors added substantial data to address concerns raised by the original reviewers. This is overall a well written and interesting study that will be of broad interest in the community.

Response to the Reviewers "Spatiotemporal dynamics of self-organized branching in pancreas-derived organoids" in Nature Communications - Second round of revisions

We are grateful to the Reviewers for their constructive comments and suggestions, which we address in our Response below, and have implemented in the updated version of the manuscript.

Note to the Reviewers: unless explicitly stated otherwise, the references given for the figures in the present response, correspond to the references of the updated manuscript. The bibliography references used in the present response, are listed and numbered in the "Bibliography" section at the end of the document.

To include the data necessary for the revisions, we have updated the Zenodo repository and its DOI. The new identifier is: <https://doi.org/10.5281/zenodo.6577226>

Reviewers can access the repository during the revision period using the following access token: https://zenodo.org/record/6577226?token=eyJhbGciOiJIUzUxMiIsImV4cCI6MTY1NjEwNzk5OSwiaWF0IjoxNjUzNDY4MTQzfQ.eyJkYXRhIjp7InJlY2lkIjo2NTc3MjI2fSwiaWQiOiJzNDE5LCJybmQiOiI5MDcwMjUxMjUzNDY4MTQzIiwiaWF0IjoxNjUzNDY4MTQzIiwiaXNjaWkiOiJkXNCJ9.3gctAc3ITyqZlpoXBZi3oLK1kugTeD3N-1TYSpX-zh90VT9zoL_yfroW_UyUQm8jyZtV_cF3qnDF2Hp1h7DB7Q

The sequencing data that support the findings of this study are deposited in the GEO database under accession code GSE200308.

Reviewers access is possible using the following token:

otulgoesrdshrkh

Reviewer #1 (Remarks to the Author):

The authors have significantly improved the manuscript and addressed all but one of my concerns.

My main concern is regarding the WT controls. The materials and methods section gives me the impression that WT cells were cultured differently (media and coated plate vs flask) compared to the mutant cells prior to being subjected to the floating collagen assay. As differences in culture conditions have an effect of cell state and can thereby affect the ability of the cells to form branching structures.

We entirely agree with reviewer that prior culture conditions can significantly affect the behavior of cells *in vitro*, in general, as well as, in particular, in our branching assay. Pancreatic ductal cells (WT PDCs) which have been isolated from normal pancreas require specific media conditions (1). Indeed, wild-type PDCs are more demanding in terms of growth factors and supplements, and cannot be cultured in the standard PDAC organoid medium (DMEM high glucose + 10% FBS + 1x P/S). Since PDCs do not grow in the PDAC organoid medium, we had previously included an image of PDCs in standard PDC culture condition in Figure 1f. As indicated in the previous response, although PDC medium allows branching in PDAC cells, we still observe more cystic structures than in the standard

media condition suggesting, indeed, a certain inhibition of branching morphogenesis. To avoid confusion for the reader, we have removed the image of the wild-type PDC in PDC medium conditions and instead show no growth under PDAC organoid media in Figure 1f. To further demonstrate that the PDCs are healthy and growing, as expected, we have included images of PDAC organoids & PDC organoids under 3 different media conditions: PDAC organoid medium, PDAC organoid medium + 20ng EGF and PDC medium in Fig. S2C.

Could the authors please clarify this. If different culture conditions are used, I believe the WT control might not fully serve its purpose. Ideally the experiment should be repeated using the same conditions for the control and mutant cells. Though, further clarification and making this obvious to the reader would also be satisfactory.

This could be achieved by:

- A schematic overview could be added in Fig1 that shows the procedures for the mutant and the wildtype cells (this would make it clear to the readers -as many do not carefully read the materials and methods section)

As requested, we have also added a schematic overview for the culture procedures of the PDC and PDAC cells in a new figure: Fig. S7. Please note also the response to the previous comment and the updated Fig. S2c.

- Materials and methods section could be changed to have one section for the healthy cells and one for the mutant cells – to make it clear to the reader that there is indeed a significant difference.

We have separated the materials and methods sections for wild-type and cancer cells in the revised version of the manuscript.

In fig1 or as a supplementary I would like to see a zoomed-out image (of a well, or significant portion of a well) containing WT organoids as well a well containing branching organoids.

This is to show that the observed phenotypic difference holds true on a population level and not only for a single organoid. to give the reader an idea what the population looks like.

This has been added to Fig. S2c

If these things are addressed, I am satisfied with the revision. The authors have greatly improved the manuscript and I appreciate the extended explanation, data and analysis with regards to their model as it highlights its robustness and applicability.

Reviewer #3 (Remarks to the Author):

In the revised manuscript the authors have extended quantitative experimental data and

computational modelling. Notably experiment and model predictions are largely in agreement in the absence of pharmacological perturbations, whereas modelling of branching in the presence of pharmacological inhibitors require further clarification and/or refinement. The authors have also improved the manuscript flow.

Specific comments:

1) Section, A minimal theoretical model captures branching organoid morphogenesis, Fig 2E, Fig 4b l. 212 – 222.

>> The author observed a negative relationship between cell proliferation rate and time as well as branch volume growth and branch width. The authors attribute these observations to negative mechanical feedback, whereas models has been also proposed to describe growth inhibition based on diffusional limitations e.g. <https://doi.org/10.1093/jb/mvu087>, <https://doi.org/10.1098/rsob.130088>. The authors should either provide convincing arguments why they ruled out diffusional limitations or refer readers to both as plausible mechanisms.

We thank the Reviewer for their comment and would like to clarify our modelling choices.

We note crucially that, as described in Fig. 4b, the relationship between growth rate and branch width, is first and foremost an experimental observation, that appears well fitted by a negative linear dependency.

This led us to introduce a general term, agnostic to the underlying molecular mechanism, for negative feedback ($-\alpha$) on thickness (through the cell density $\rho(t)$) in the conservation equation for cell number, such that $N(\dot{t}) = k_d N(\dot{t})(1 - \alpha \rho(\dot{t}))$.

Previously published papers have proposed that such feedbacks could originate from stress or density effects on proliferation (see (2–6) for already cited references in the manuscript, or for instance (7–11), for additional references)

Importantly however, we did not explicitly rule out any model, and considered mechanical effects as a plausible, well-documented hypothesis, for the origin of the observed dependency. For completeness, we have referred the reader to other mechanisms in the manuscript section “A minimal theoretical model captures branching organoid morphogenesis”.

2) L 251. “see Supplementary Note for details, and Fig. 4d-h, Fig. S5 for typical simulation outputs”

>> Fig. S5 does not contain any simulation output

We thank the Reviewer for catching the typo, and have corrected it to Fig. S6 in the manuscript.

3) L 255 – 257. “Secondly, our model also predicted that the variability and variance in branch number across organoids should also grow exponentially, and be of the same order of magnitude as the average itself – characteristic of a stochastic branching process. Strikingly, this described very well the data (Fig. 4g), arguing that most of the heterogeneity in branching organoid size could possibly arise from the intrinsic stochasticity of branching, rather than intrinsically different average rates of branching.”

>> proportional error is characteristic for many processes. If authors would like to rule out a

particular mechanism based, it's incompatibility with the variance to the mean must be explicitly demonstrated.

We apologise for the lack of clarity. We first note that we replaced the term “variance” by “standard deviation” for consistency with the plot in Fig. 4g. Secondly, we here we meant not only that proportional error could be obtained by stochasticity of the branching process, but that the magnitude of the variability could already be explained by it (as the standard deviation could have scaled in the same way but with a weaker prefactor). Importantly, we do not exclude other contributions to the variability across organoids: however, these contributions are expected to be additive, arguing that most of the heterogeneity in branching organoid size could already arise from the intrinsic stochasticity of branching, rather than intrinsically different average rates of branching.

To further strengthen our point, we have investigated the distribution of non-terminal branch lengths (i.e. the length of formed segments between two branching points). Our model, and this explanation for organoid variability, relies on a stochastic model of branching, therefore predicting broad exponential distributions for the branch length. Indeed, plotting non-terminal branch length distributions at day 7-10 revealed broad distributions lacking a well-defined peak. We have added this plot as an updated Fig. S5e and have added this rationale to the section “A minimal theoretical model captures branching organoid morphogenesis” of the manuscript.

4) Fig. 4e.

>>Experimental data suggests that the average branch thickness continues to increase over 9 day period, whereas the model predicts flattening of the average thickness after 4 days. A discussion of this apparent discrepancy will benefit the reader.

The deviation of the “experimental” terminal branch average thickness from the “theoretical” value, at day 8 and 9 can be explained by the fact that, due to heterogeneity, some organoids phenotypes can display budding and occasionally start lumen expansion earlier than others, thereby driving the increase in apparent average thickness. As our described model does not take into account the lumen formation mechanisms, for reasons described in the previous letter and in the supplementary text, such discrepancies might emerge.

While, for consistency, we have taken care to exclude branches that displayed an obvious, large and optically resolvable lumen, early lumen initiation (which can still lead to a noticeable thickness increase) in branch tips can be difficult to detect using the method described in the Materials and Methods section “Branch thickness measurement (static)”.

We have added below (see Fig. R1 of the current letter) images of organoids at day 9, cultured in the same well, to illustrate this point.

Figure R1 - Phenotypic variability among day 9 organoids cultured in the same well. The left organoid is still at a stage of development with thin spiky tips, whereas the right organoid displays rounded tips. The red arrow indicates a clear, optically resolved lumen in the core of a branch; the magenta arrow indicates an optically resolvable lumen in the core of a branch, but where the bud tip still appears “closed” and therefore has its thickness measured; the cyan arrows indicate an area where it is unclear whether a lumen has already opened, therefore the tip thickness on the corresponding branch was measured. Pictures are summed slices projections of organoids’ cell membranes stained with 0.05% CellMask, imaged with a confocal microscope. Scale bars: 200 μm .

We have added a discussion of this point to the supplementary text, in section II. C - “Predictions from the model on unperturbed and perturbed organoid growth”.

5) L 269: “Finally, the model can be tested by observing the influence of batimastat treatment (a matrix metalloproteinase inhibitor, further described in the following section), which was found to abolish tip invasion speed ($v_0 \approx 0$). While branch volumetric growth is also perturbed in this condition (-77%), it remains active to a degree, so that by simple conservation law, we predict it to cause marked thickening of organoids (Fig. S6b-c), which was observed experimentally (Fig. 5c, Fig. S6f), as described above.”

>> Model sensitivity analysis indicates that the invasion speed has virtually no effect branch thickness (Fig Sup 6 b). This is contrary to the detailed experimental observation in presence of MMP inhibitors (Fig. S4)

We apologize for the confusion: v_0 does have some effect on the branch thickness. When organoids branch and elongate, the thickness converges towards a steady state value that is smaller than the “maximal thickness” of growth arrest (in our data, 20 microns instead of 25 microns). In Fig. S6b, we considered smaller changes to v_0 (specifically dividing v_0 by 2), whereas batimastat caused an on/off switch – reducing v_0 to 0, and thus a larger effect. We clarified this in the main text.

6) L. 274: “However, if proliferation is inhibited, as with an aphidicolin treatment (a proliferation inhibitor, further described in the following section), the model predicts thinning of the organoid in time.”

>> Are these modelling results explicitly depicted in the publication?

This was in Fig. S6d: changing cell division changes the steady-state thickness of the organoids – thinning in this case. We have added the figure reference to the main text.

Reviewer #4 (Remarks to the Author): Expert in pancreatic cancer and tumour microenvironment

The current manuscript by Dr Bausch and colleagues described a new method to self organizing, branching organoids that replicate the structure of pancreatic cancer; the authors define different phases of the organoid organization process, each defined by its own gene expression pattern. Further, the authors developed a model to describe the process of branching morphogenesis in the organoids. Overall, the authors developed a new #D model to mimic growth and 3D organization of pancreatic cancer, complementing existing models in the field.

In this revised application, the authors added substantial data to address concerns raised by the original reviewers. This is overall a well written and interesting study that will be of broad interest in the community.

We thank the Reviewer for their report.

Bibliography

1. M. Reichert, S. Takano, S. Heeg, B. Bakir, G. P. Botta, A. K. Rustgi, Isolation, culture and genetic manipulation of mouse pancreatic ductal cells. *Nat. Protoc.* **8**, 1354–1365 (2013).
2. B. I. Shraiman, Mechanical feedback as a possible regulator of tissue growth. *Proc. Natl. Acad. Sci. U. S. A.* **102**, 3318–3323 (2005).
3. F. Montel, M. Delarue, J. Elgeti, L. Malaquin, M. Basan, T. Risler, B. Cabane, D. Vignjevic, J. Prost, G. Cappello, J. F. Joanny, Stress clamp experiments on multicellular tumor spheroids. *Phys. Rev. Lett.* **107**, 1–4 (2011).
4. M. Delarue, F. Montel, D. Vignjevic, J. Prost, J.-F. Joanny, G. Cappello, Compressive Stress Inhibits Proliferation in Tumor Spheroids through a Volume Limitation. *Biophys. J.* **107**, 1821–1828 (2014).
5. M. Delarue, J. Hartung, C. Schreck, P. Gniewek, L. Hu, S. Herminghaus, O. Hallatschek, Self-

- driven jamming in growing microbial populations. *Nat. Phys.* **12**, 762–766 (2016).
6. S. J. Streichan, C. R. Hoerner, T. Schneidt, D. Holzer, L. Hufnagel, Spatial constraints control cell proliferation in tissues. *Proc. Natl. Acad. Sci.* **111**, 5586–5591 (2014).
 7. T. Aegerter-wilmsen, M. B. Heimlicher, A. C. Smith, P. B. De Reuille, R. S. Smith, C. M. Aegerter, K. Basler, Integrating force-sensing and signaling pathways in a model for the regulation of wing imaginal disc size. **3231**, 3221–3231 (2012).
 8. L. LeGoff, H. Rouault, T. Lecuit, A global pattern of mechanical stress polarizes cell divisions and cell shape in the growing *Drosophila* wing disc. *Development*. **140**, 4051–4059 (2013).
 9. Y. Pan, H. Alégot, C. Rauskolb, K. D. Irvine, The dynamics of Hippo signaling during *Drosophila* wing development. *Development*. **145**, dev165712 (2018).
 10. T. Hirashima, T. Adachi, Polarized cellular mechano-response system for maintaining radial size in developing epithelial tubes. **2** (2019), doi:10.1242/dev.181206.
 11. T. Hirashima, Mechanical Feedback Control for Multicellular Tissue Size Maintenance : A Minireview. **9**, 1–6 (2022).
 12. T. Miura, K. Shiota, Depletion of FGF acts as a lateral inhibitory factor in lung branching morphogenesis in vitro. *Mech. Dev.* **116**, 29–38 (2002).
 13. C. Greggio, F. De Franceschi, M. Figueiredo-Larsen, S. Gobaa, A. Ranga, H. Semb, M. Lutolf, A. Grapin-Botton, Artificial three-dimensional niches deconstruct pancreas development in vitro. *Development*. **140**, 4452–4462 (2013).
 14. T. Miura, Models of lung branching morphogenesis. *J. Biochem.* **157**, 121–127 (2015).
 15. S. B. Dahl-Jensen, M. Figueiredo-Larsen, A. Grapin-Botton, K. Sneppen, Short-range growth inhibitory signals from the epithelium can drive non-stereotypic branching in the pancreas. *Phys. Biol.* **13** (2016), doi:10.1088/1478-3975/13/1/016007.

REVIEWERS' COMMENTS

Reviewer #1 (Remarks to the Author):

In the second round of revision the authors of the manuscript 'Spatiotemporal dynamics of self-organized branching in pancreas-derived organoids' have addressed all my comments. The manuscript is well written and the authors have added a significant amount of data which improves the quality of the manuscript.

Reviewer #3 (Remarks to the Author):

In the revised version of the manuscript the authors have addressed my earlier comments. However, the authors have introduced additional statements, which either require further clarification or perhaps could be omitted.

Lines 266 – 270: "Thirdly, as our model – and this explanation for the organoid variability – relies on a stochastic model of branching, we predicted broad, exponential distributions for branch length itself (rather than the tightly peaked distributions around a given value that would be expected for a mechanism such as Turing instabilities 22)."

Broad distribution of the branching length is likely the result of the continuous branch growth, also after the branching event. If authors would like to make a statement regarding the impact of the mechanism on the branch length distribution, the analysis should be stratified by the branch generation or branch "age" (<http://dx.doi.org/10.1088/1478-3975/9/6/066006>). Turing instabilities are indeed known to produce patterns with a given wave length on simple domains, however on the complex 3D domains pattern wavelength is modulated by the local geometry and environment via diffusional fluxes.

Response to the Reviewers "Spatiotemporal dynamics of self-organized branching in pancreas-derived organoids" in Nature Communications

Reviewer #1 (Remarks to the Author):

In the second round of revision the authors of the manuscript 'Spatiotemporal dynamics of self-organized branching in pancreas-derived organoids' have addressed all my comments. The manuscript is well written and the authors have added a significant amount of data which improves the quality of the manuscript.

We thank the Reviewer for their positive assessment.

Reviewer #3 (Remarks to the Author):

In the revised version of the manuscript the authors have addressed my earlier comments. However, the authors have introduced additional statements, which either require further clarification or perhaps could be omitted.

Lines 266 – 270: "Thirdly, as our model – and this explanation for the organoid variability – relies on a stochastic model of branching, we predicted broad, exponential distributions for branch length itself (rather than the tightly peaked distributions around a given value that would be expected for a mechanism such as Turing instabilities 22)."

Broad distribution of the branching length is likely the result of the continuous branch growth, also after the branching event. If authors would like to make a statement regarding the impact of the mechanism on the branch length distribution, the analysis should be stratified by the branch generation or branch "age" (<http://dx.doi.org/10.1088/1478-3975/9/6/066006>). Turing instabilities are indeed known to produce patterns with a given wave length on simple domains, however on the complex 3D domains pattern wavelength is modulated by the local geometry and environment via diffusional fluxes.

We have clarified that Turing instabilities *on a simple domain* would lead to periodic branching and thus a highly peaked distribution for the branch length, while recognizing the possibility that continuous ductal growth could cause more complex dependencies. In line with the Reviewer's recommendations, we have indicated that measuring in detail the relationship between branch age and length would be an insightful future experiment.